

# Effects of mud supply on large-scale estuary morphology and development over centuries to millennia

Lisanne Braat[1], Thijs van Kessel[2], Jasper R.F.W. Leuven[1], and Maarten G. Kleinhans[1]

[1]Utrecht University, Heidelberglaan 2, 3584 CS Utrecht, the Netherlands
[2]Deltares, Boussinesqweg 1, 2629 HV Delft, the Netherlands

*Correspondence to:* Lisanne Braat (L.Braat@uu.nl)

**Abstract.** Alluvial river estuaries consist largely of sand but are typically flanked by mud flats and salt marshes. The analogy with meandering rivers, that are kept narrower than braided rivers by cohesive floodplain formation, raises the question how large-scale estuarine morphology and late Holocene development of estuaries are affected by cohesive sediment. In this study we combine sand and mud transport processes and study their interaction effects on morphologically modelled estuaires on centennial to millennial time-scales. The numerical modelling package Delft3D was applied in 2DH starting from an idealised convergent estuary. The mixed sediment was modelled with an active layer and storage module with fluxes predicted by the Partheniades-Krone relations for mud, and Engelund-Hansen for sand. The model was subjected to a range of idealised boundary conditions of tidal range, river discharge, waves and mud input. The model results show that mud is predominantly stored in mudflats on the side of the estuary. Marine mud supply only influences the mouth of the estuary whereas fluvial mud is distributed along the whole estuary. Coastal waves stir up mud and remove the tendency to form muddy coastlines and the formation of mudflats in the downstream part of the estuary. Widening continues in estuaries with only sand while mud supply leads to a narrower constant width and reduced channel and bar dynamics. This self-confinement eventually leads to a dynamic equilibrium where lateral channel migration and mud flat expansion are on average balanced. However, for higher mud concentrations, higher discharge and low tidal amplitude the estuary narrows and fills to become a tidal delta.

## 1 Introduction

River estuaries with continuously migrating channels and bars have great and often conflicting economic and ecologic values. These estuaries are typically dominantly built of sand, but mud and salt marshes also form significant parts of these systems. Mud plays a critical role in ecological restoration measures and harbour maintenance, but is rarely taken into account in numerical morphological models. Due to human interference mud concentrations have increased far above the desired values in many estuaries (Winterwerp, 2011; van Maren et al., 2016). Mud problems arise from pollutants attached to clay particles, mud deposits covering benthic species, rapidly silting harbours and channels and changing hydro- and morphodynamic conditions by higher resistance against erosion. This raises questions about effects of mud on large-scale estuary morphology in natural alluvial systems as a control for cases with human interference.





In rivers, formation of cohesive floodplains with mud and vegetation causes river channels to be narrower and deeper than in systems with only sand given otherwise the same conditions (Tal and Paola, 2007; Kleinhans, 2010; van Dijk et al., 2013; Schuurman et al., 2016). This results from a dynamic balance between floodplain erosion by migration of channels and new floodplain formation by mud sedimentation and/or vegetation development. The effective cohesiveness may change an uncon-

fined braided system into a dynamic self-confined meandering system or even a straight, laterally immobile channel without bars (Makaske et al., 2002; Kleinhans and van den Berg, 2011). Here we study whether mud has similar effects on large-scale planform that develop over centuries to millennia in estuaries. We especially need more knowledge about where mud deposits occur and how they influence the evolution of the estuary over long timescales. We first quantify mud flat properties in two Dutch estuaries and then review approaches to mud modelling.

## 1.1 Spatial pattern of mud flats in estuaries

In this study we use data from two Dutch estuaries, the Western Scheldt estuary and the Eems-Dollard estuary, because they are relatively well documented, although bed composition data is rather scarce compared to bed elevation scans. The disadvantage of data in a well-studied estuary is that anthropogenic influences are usually considerable, so we only look at general patterns and properties. Here we combine independent measures for mud in the bed: 1) a bed sampling dataset of the Western Scheldt

(Fig. 1a; McLaren, 1993, 1994), 2) probability of clay in the GeoTOP map (v1.3) of interpolated borehole data in the top $50 \, \mathrm{cm}$ of the bed (TNO, 2016) (Fig. 1b and 1e), where clay is defined as more than $35 \, \%$ lutum ($< 2 \, \mu\mathrm{m}$) and less than $65 \, \%$ silt ($< 63 \, \mu\mathrm{m}$) (Vernes and van Doorn, 2005), 3) yearly Western Scheldt ecotope maps of Rijkswaterstaat (2012), in particular the mud-rich areas above low water level (Fig. 1c), that are based on aerial photographs, and 4) the sediment atlas of the Waddensee (Rijkswaterstaat, 2009) drawn from bed sampling in 1989 (van Heuvel, 1991) which includes the Eems-Dollard (Fig. 1d).

Data from the two estuaries indicate that mud is deposited on the sides of the estuary (Fig. 1a–d) and larger fractions of mud are especially found in areas on the sides and bars shielded from the strongest tidal flow. The hypsometric curves (Fig. 1g and h) indicate that most of the mud is deposited on the intertidal area, yet, significant fractions are also found in channels. Additionally larger mud fractions are observed in the single-channel upper estuaries (Fig. 1a, d and f). To summarise, $10 - 20 \, \%$ of the lower estuary cross-section is typically covered by mud, increasing to higher fractions up to about half the cross-section

in the single-channel upper estuary.



**Figure 1.** Mud in the bed of the Western Scheldt and the Eems-Dollard. (a) Percentage of mud in the top 10 cm of the bed (McLaren, 1993, 1994), (b) GeoTOP map (v1.3) of probability of clay in the top 50 cm of the bed (TNO, 2016), (c) indicative morphodynamics map of the Western Scheldt (Rijkswaterstaat, 2012). (d) Fraction of mud in the top 10 cm of the bed (van Heuvel, 1991; Rijkswaterstaat, 2009), (e) GeoTOP map (v1.3) of probability of clay occurrence in the top 50 cm of the bed (TNO, 2016). (f) Surface mud distribution along the Western Scheldt from the three datasets. For the ecotope data only the low dynamics muddy class was used. (g) and (h) Cumulative and normalised hypsometric curves of surface area related to bed elevation. Plot includes the (cumulative and normalised) distribution of mud relative to the total area with reference to figure panel for the mud datasets. Dotted lines indicate high and low water levels during spring and neap tide at the mouth.



## 1.2 Past and novel modelling approaches for sand-mud mixtures

In past modelling, sand and mud were always considered separately, partly because the interactions between sand and mud are complicated. Models used either sand (e.g., van der Wegen et al., 2008) or sand and mud without interactive transport (e.g., Sanford, 2008). However, there is interaction between sand and mud that affects the erodibility (see van Ledden et al., 2004a, for review). Such interactions include that dominant mud with some sand behaves as mud, but for lower mud fractions there is mixed behaviour (van Ledden et al., 2004a). In particular, mixed sediments increase erosion resistance and decrease erosion rates when the critical shear stress is exceeded, compared to pure sand (e.g., Torfs, 1995; Mitchener and Torfs, 1996). This behaviour is highly sensitive to small amounts of mud, and the highest critical shear stresses for erosion occur with $30-50 \, \text{wt} \, \%$ sand (e.g., Mitchener and Torfs, 1996).

Over the past decade, mixed sediments were implemented in several modelling software packages (van Ledden et al., 2004a; Waeles et al., 2007; van Kessel et al., 2011; Le Hir et al., 2011; Dam et al., 2016). Long-term morphologic calculations are rare due to computer limitations and lack of spatially and temporally dense data of mud in the bed. For deltas on the other hand, long-term morphologic development by numerical modelling (Edmonds and Slingerland, 2009; Caldwell and Edmonds, 2014; Burpee et al., 2015) showed large effects of mud on plan-shapes, patterns and dynamics with fairly simplistic sediment transport processes. In particular, cohesion reduces the ability to re-erode, resulting in more stable bars and levees, longer and deeper channels, less active channels and less channel migration. Similar results were found in physical experiments of deltas (Hoyal and Sheets, 2009) and of river meandering (van Dijk et al., 2013). However, the sensitivity of the numerical models to parameters such as erodibility and settling velocity indicate that the value of long-term modelling exercises with the current state of the art is to develop generalisations and trends rather than precise hindcasts and predictions of specific cases.

Past long-term morphological modelling studies of estuaries that did not include mud, showed that channel-bar patterns form that are similar to those in nature (Hibma et al., 2003; van der Wegen and Roelvink, 2008; van der Wegen et al., 2008; Dam et al., 2013). Cases where boundaries eroded unhindered (van der Wegen et al., 2008) developed towards a state of decreasing morphodynamic activity as size and depth continued to increase and morphodynamic equilibrium was not reached. Most models, however, including the few models with mud, assumed prescribed planform shapes with unerodible boundaries (Lanzoni and Seminara, 2002; Hibma et al., 2003; van der Wegen and Roelvink, 2008; Dam et al., 2013; Dam and Bliek, 2013) allowing equilibrium in some cases. However, to obtain a dynamic equilibrium of planform shape and dimensions, where on average bank erosion equals sedimentation, the formation of cohesive mud flats needs to be incorporated in models with erodible banks. Regardless of the fact that most natural estuaries are in disequilibrium as they continuously adapt to changing boundary conditions and anthropogenic influences, it is of interest to know whether these systems could develop a morphodynamic equilibrium and on which variables this depends most.

The objective of this research is to determine the effect of mud supply on equilibrium estuary shape and dynamics. This fills a gap in literature by combining long-term morphological modelling of both estuaries and the effect of sand-mud interaction. We examine estuary formation from idealised initial conditions and a range of boundary conditions and run models for $2000 \, \text{yr}$ in order to study tendencies towards dynamic equilibrium. We hypothesise that mud will settle into mud flats flanking the



estuary that resist erosion and thus self-confine and narrow the estuary and reduce channel-bar mobility and braiding index. As a result we expect that self-formed estuaries develop a dynamic balance between bank erosion on the one hand and bar and mud flat sedimentation with resistant cohesive mud on the other hand.

## 2 Methods

The methodology was to set up an idealised scenario loosely inspired by the Dyfi, i.e. Dovey, estuary in Wales, and to vary the most relevant boundary conditions for the main question. These include mud concentration supplied at the upstream boundary, mud supplied at the coastal boundary, surface waves, river discharge and tidal amplitude. There is a host of other initial conditions, boundary conditions and other variables that can be tested such as other tidal components and other initial valley shapes. For example, application of certain tidal components can lead to change import or export tendencies of tidal systems, as can river inflow (Guo et al., 2016). However, our aim is to isolate effects of mud which requires the simplest possible conditions without non-linear interactions between imposed tidal components. Furthermore, we tentatively assume that the model is sufficiently sophisticated to reproduce the general behaviour found in nature of the phenomena under investigation, which will be discussed later.

### 2.1 Numerical model description

This numerical modelling study used the modelling package DELFT3D version 4.01.00, which is a process-based simulation program and consists of several integrated modules (Lesser et al., 2004). This model is state-of-the art, open source, widely used and tested, and includes the possibility to use both sand and mud in the calculations. The depth averaged version of DELFT3D was used to keep the computational time for long-term simulations below a month. Furthermore, we excluded the effect of the salinity and temperature on the hydrodynamics, as it was assumed that the effect of density differences would be limited in 2DH and in well-mixed shallow estuaries. Auxiliary tests in 3D with 5 layers and salinity confirm the assumption of well-mixed conditions. Furthermore, the Estuary-Richardson number (Fischer, 1972, as defined by) is 0.036 and the Rouse number is $< 0.01$, further supporting the assumption of a well-mixed estuary for salinity and suspended sediment. Effects of the Coriolis force, organisms and wind are ignored for generalisation and simplicity.

Hydrodynamics were calculated by solving the Navier-Stokes equations for incompressible fluid with Boussinesq and shallow water assumptions. This includes the continuity equation:

$$\frac{\partial \eta}{\partial t} + \frac{\partial hu}{\partial x} + \frac{\partial hv}{\partial y} = 0 \tag{1}$$

and the momentum equations:

$$\frac{\partial u}{\partial t} + u\frac{\partial u}{\partial x} + v\frac{\partial u}{\partial y} + g\frac{\partial \eta}{\partial x} + \frac{gu\sqrt{u^2+v^2}}{(C^2 h)} - v_w\left(\frac{\partial^2 u}{\partial x^2} + \frac{\partial^2 u}{\partial y^2}\right) = 0 \tag{2}$$

$$\frac{\partial v}{\partial t} + u\frac{\partial v}{\partial x} + v\frac{\partial v}{\partial y} + g\frac{\partial \eta}{\partial x} + \frac{gv\sqrt{u^2+v^2}}{(C^2 h)} - v_w(\frac{\partial^2 v}{\partial x^2} + \frac{\partial^2 v}{\partial y^2}) = 0 \tag{3}$$



**Table 1.** Sediment characteristics applied in the default model. Sensitive parameters will be varied in scenarios discussed later.

| Sediment property | symbol | value | unit |
|---|---|---|---|
| Sand | | | |
| Median grain size | $D_{50}$ | $3 \times 10^{-4}$ | $m$ |
| Specific density | $\rho_s$ | 2650 | $kg/m^3$ |
| Dry bed density | $\rho_{dry}$ | 1600 | $kg/m^3$ |
| Mud | | | |
| Settling velocity | $w_s$ | $2.5 \times 10^{-4}$ | $m/s$ |
| Critical bed shear stress for sedimentation | $\tau_{crit,sed}$ | 1000 | $N/m^2$ |
| Critical bed shear stress for erosion | $\tau_{crit,ero}$ | 0.2 | $N/m^2$ |
| Erosion parameter | $M$ | $1 \times 10^{-4}$ | $kg/m^2/s$ |
| Specific density | $\rho_s$ | 2650 | $kg/m^3$ |
| Dry bed density | $\rho_{dry}$ | 1600 | $kg/m^3$ |

where $\eta$ is water level with respect to datum (m), $h$ is water depth (m), $u$ is depth averaged velocity in $x$ direction (ms$^{-1}$), $v$ is depth averaged velocity in $y$ direction (ms$^{-1}$), $g$ is gravitational acceleration (ms$^{-2}$), $C$ is the Chezy friction parameter (m$^{0.5}$s$^{-1}$) and $v_w$ is the eddy viscosity (m$^2$s$^{-1}$).

A module was recently developed for mixed sediments which incorporates the effect of bed composition on erosional
behaviour and hence morphology (van Kessel et al., 2011, 2012). This module is a partial implementation of van Ledden (2001); Jacobs et al. (2011) and tracks spatial and temporal bed composition for multiple grain sizes of sand and mud with erosional characteristics depending on bed composition. In this paper we only used one sand fraction and one mud fraction (Table 1) and applied a uniform roughness.

Cohesive sediment, i.e. mud, is defined as the mixture of the clay ($< 2\,\mu$m) and silt ($2 - 63\,\mu$m) fractions, where it's cohesive
behaviour is mainly caused by physico-chemical forces between the clay particles. This cohesive behaviour causes complex processes that influence erosion and deposition of sediments. In the model we distinguish two erosion modes. Above a critical mud content ($p_{m,cr}$) of the bed, sand particles are not in direct contact but are covered by cohesive particles, which limits erosion for both sand and mud (Torfs, 1995, 1996). Below this critical mud content, friction and gravity oppose sediment transport for sand. The critical mud content was chosen to be at a mass fraction of $0.4$, which depends on site specific silt-clay
ratios because only the clay fraction is cohesive (McAnally and Mehta, 2001; van Ledden et al., 2004a). This value is higher than found in flume experiments ($0.1 - 0.2$, Torfs 1995; $0.05 - 0.15$, Torfs 1996; $0.02 - 0.15$, Mitchener and Torfs 1996),but was based on silt-clay ratios of Dutch tidal systems ($0.25 - 0.5$ van Ledden et al., 2004b).





When the bed is defined as non-cohesive ($p_m < p_{m,cr}$), a traditional sand transport equation was used. Here we chose the Engelund and Hansen transport equation (1967; Eq. 4):

$$q_s = \frac{0.05U^5}{\sqrt{g}C^3\Delta^2 D_{50}} \tag{4}$$

where $q_s$ is sediment transport ($\mathrm{m^3 m^{-1} s^{-1}}$), $U$ is the magnitude of the flow velocity ($\mathrm{ms^{-1}}$), $\Delta$ is the relative density ($\rho_s -$

$\rho_w)/\rho_w$ and $D_{50}$ is the median grain size (m). This equation does not distinguish between suspended and bedload transport, but considers total transport.

The erosion rate of mud was calculated by the Partheniades-Krone formulation (Partheniades, 1965, Eq. 5):

$$E_m = MS(\tau_{cw}, \tau_{cr,e}) \tag{5}$$

where $E_m$ is the erosion flux of mud ($\mathrm{kgm^{-2}s^{-1}}$), $M$ is the erosion parameter ($\mathrm{kgm^{-2}s^{-1}}$), $S$ is the erosion or depositional

step function, $\tau_{cr,e}$ is critical shear stress for erosion ($\mathrm{Nm^{-2}}$), and $\tau_{cr,d}$ is the critical shear stress for deposition ($\mathrm{Nm^{-2}}$).

When the bed is cohesive ($p_m > p_{m,cr}$), the mud and sand fluxes are proportional to the mud and sand fraction. The erosion rate of mud is calculated by the Partheniades-Krone formulation (Partheniades 1965; Eq. 5) similar to the non-cohesive regime. The erosion rate for sand on the other hand was based on the entrainment of mud, because sand particles are included in the cohesive matrix (Eq. 6). In this way sand can only be eroded when mud is eroded. Bed load transport was assumed to be zero

in the cohesive regime.

$$E_s = E_m \tag{6}$$

Sediment suspended following the Partheniades-Krone formulation was further described by the advection-diffusion equation. Sand and mud behave independently in suspension and segregation will occur with low concentrations (Torfs, 1996). For simplicity we assumed a constant settling velocity of 0.25 mm/s for mud, ignoring that settling velocity depends on flocculation

influenced by concentration, residence time, salinity, pH, turbulence and biochemical effects (e.g., Mietta et al., 2009). The settling velocity is typical for fluvial mud ($0.1 - -0.4\ \mathrm{mms^{-1}}$ Temmerman et al., 2003), which we supply in our default run, and is relatively low for marine mud.

Bed level change was determined by the gradients in sediment transport. To track the mud and sand fractions in the bed, a bed module was used with a mixed Eulerian-Lagrangian approach (van Kessel et al., 2011, 2012) similar to Le Hir et al. (2011);

Sanford (2008). An active layer of 10 cm was used where sediment exchange occurs with the water column. This active layer had a constant thickness and moved through the vertical framework with bed aggradation and degradation. Below the active layer we used several Eulerian layers to store bed composition in the vertical (Table 2). The advantage of Eulerian bed-layers is that numerical diffusion is minimal while a Lagrangian active layer maintains constant thickness so that the calculation time is more stable and strong bed armouring is prevented.

To speed up morphodynamic calculations the bed level change in each time step was multiplied with a morphological factor of 400 (Table 2). Extensive studies showed reasonable results up to a morphological factor of 1000, though it is recommended for Delft3D not to go over 400 (Roelvink, 2006; van der Wegen and Roelvink, 2008). Using a morphological factor is an



**Table 2.** Parameters for processes and numerics

| Parameter | symbol | unit | value |
|---|---|---|---|
| Time step | $dt$ | min | 0.3 |
| Spin up time at cold start | – | min | $1.44 \times 10^4$ |
| Threshold depth drying/flooding | – | m | 0.08 |
| Min water depth for bed level change | – | m | 0.05 |
| Erosion adjacent dry cells | – | – | 0.5 |
| Morphological factor | $Morfac$ | – | 400 |
| Transverse bed slope parameter | $\alpha$ | – | 0.2 |
| Transverse bed slope parameter | $\beta$ | – | 0.5 |
| Eulerian bed storage layer thickness | – | m | 0.1 |
| Active layer thickness | – | m | 0.1 |

efficient way of speeding up long-term morphodynamic calculations that is widely used (Roelvink, 2006; van der Wegen and Roelvink, 2008; Le Hir et al., 2015; Dam et al., 2016).

When the water level changes during a tidal cycle, flooding and drying of intertidal area occurs. To prevent complicated and time-consuming hydrodynamic calculations with very small water depths a threshold is set for drying and flooding (Table 2).

When the water depth is below this threshold the velocity is set to zero. Since the velocity in dry cells is zero, there is no sediment transport in dry cells, even when considerable erosion occurs in a wet cell next to it. Therefore, dry beach and bank erosion was implemented to drive lateral bed lowering. A user-defined factor (Table 2) determines the fraction of the erosion flux that is assigned to the adjacent dry cells.

The transverse bed slope effect is a very important parameter for bar dimensions and behaviour in morphological models

(Schuurman et al., 2013) that is often used as a calibration parameter. In estuary models the transverse bed slope effect is often set to be much stronger than the advised default settings to prevent unrealistically steep banks and narrow bars and channels (van der Wegen and Roelvink, 2012). The reason for this is unclear but unravelling this is beyond the scope of the present paper so we use settings similar to earlier studies (van der Wegen and Roelvink, 2012). We used the transverse bed slope predictor of Koch and Flokstra (1980) as extended by Talmon et al. (1995):

$$f(\theta) = \alpha \theta^\beta \qquad (7)$$

where $\theta$ is the shields parameter, $D$ median grain size (m), $H$ the water depth (m) and $\alpha = 0.2$, much lower than the default of 1.5 for rivers, and $\beta = 0.5$.

The Engelund-Hansen transport formulation was chosen because other relations, in particular van Rijn (1993); van Rijn et al. (2004); van Rijn (2007), resulted in higher bars and much deeper and straight channels with sudden (up to 90 °) sharp bends,

which would require transverse bedslope parameters that differ two orders of magnitude from the theoretical value in estuarine settings (van der Wegen and Roelvink, 2012). Furthermore, changing bedslope parameters does not fix the channel pattern



issues. For long-term morphological modelling Engelund-Hansen produces more realistic morphologies. The disadvantage of our method is that the present code for sand-mud interaction with Engelund-Hansen does not yet incorporate a gradual transition in critical shear stress for erosion between de cohesive and non-cohesive regime. Additionally, mud would ideally erode proportionally with sand in the non-cohesive regime as sand erodes with mud in the cohesive regime, but this is not

yet implemented for Engelund-Hansen and is therefore also not described in our method section. These issues are beyond the scope of the present paper and require further research and model code development.

## 2.2 Model schematization

The modelled domain is 30 by 15 km of which 10 by 15 km is sea area (Fig. 2). The grid is rectilinear with a non-uniform resolution. Cell size varies between 50 by 80 and 125 by 230 m, and increases in size from the initial estuary shape to the

sides and offshore to increase resolution in regions of interest and to decrease computation time. The initial bathymetry is in the shape of an idealised funnel-shaped estuary. This exponential shape is also found in previous modelling research (Lanzoni and Seminara, 2002; Canestrelli et al., 2008; Lanzoni and D'Alpaos, 2015) and obtained from field data (Savenije, 2015). The estuary is 3 km at the mouth and decreases exponentially to a channel of 300 m wide over 20 km. The bed level linearly increases in elevation from $-2$ at the mouth to 2 m at the upstream boundary and 2 to 3 m on dry land (van der Wegen and

Roelvink, 2008). The sea has an depth of 15 m. van der Wegen and Roelvink (2008) argued that initial bathymetry does not greatly affect the dynamic equilibrium shape, because dry-cell or bank erosion is allowed in the model and the model will therefore develop a self-formed (alluvial) estuary shape. However, initial shape affects the time needed to form the equilibrium planform shape as well as the size of the ebb delta in the absence of waves and littoral transport, which is the default situation in our idealised estuary. We therefore started with a funnel shape to save calculation time and decrease the size of the ebb tidal

delta. The shape is given as:

$$W = W_{mouth} e^{\left(\frac{-x}{L_b}\right)} \qquad (8)$$

where $W_{mouth} = 3000$ m is the width of the estuary at the mouth , $L_b = 3362.6$ m is the e-folding distance over which the width of an exponential channel is reduced by a factor of $e$, and $x$ is distance from the mouth (m). The shapes of modelled estuaries are characterised by the funnel shape parameter (Davies and Woodroffe, 2010) calculated as e-folding length norma-

lised by mouth width at that point in time (Eq. 9). Lower values of the characteristic funnel length indicate stronger funnelling in the sense of more rapid narrowing from the mouth in landward direction. In this way estuary shape is normalised by estuary size.

$$S_b = L_b / W_{mouth} \qquad (9)$$

Three open boundaries are used: two cross-shore water level boundaries and one upstream discharge boundary. At the water

level boundaries an M2 tide is prescribed with a default tidal amplitude of 1.5 m and a phase difference of 3 ° (6 min over 15 m). With these simple conditions tidal asymmetry in the estuary is entirely caused by basin geometry and river flow and not by prescribed overtides. For generalisation purposes and simplicity we exclude known effects of imposed multiple tidal




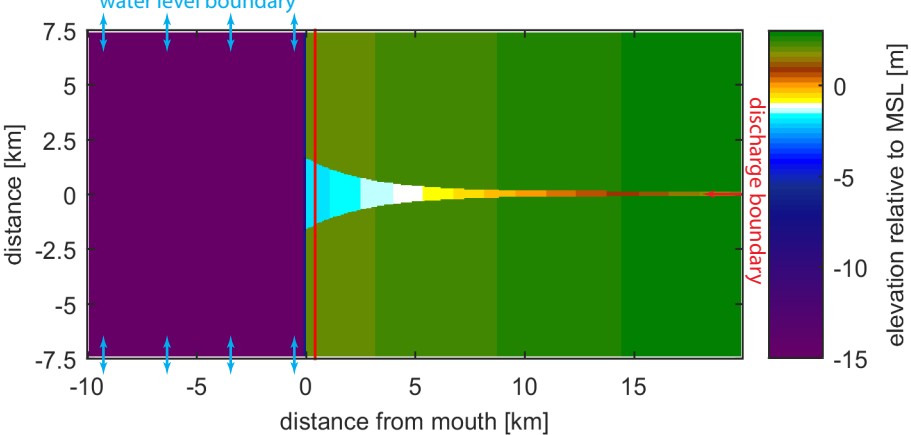

**Figure 2.** Initial bathymetry with model boundaries and cross-section (red line) for analysis. Initial depth increases linearly upstream and width decreases exponentially (Eq. 8). Coordinates are defined at the coastline with the channel centreline and mean sea level (MSL) as origin.

constituents on residual transport and morphology (Guo et al., 2016). Discharge is prescribed as a constant value through time of $100 \ \mathrm{m^{-3}s^{-1}}$ over the inflow cross-section. However, the partitioning of discharge between the upstream grid cells varies sinusoidally through time from one side to the other to simulate weak upstream 'meandering' perturbations with a period of $500 \ \mathrm{yr}$ to trigger bars if the self-formed channel aspect ratio would allow bars (Schuurman et al., 2016).

5     In some model scenarios waves were imposed at all sea boundaries including the closed, offshore boundary parallel to the coast. Waves have a $6 \ \mathrm{s}$ peak period and a significant wave height of $0.7 \ \mathrm{m}$. This is the highest possible significant wave height for which no coastal erosion occurs. This allows us to isolate the effect of waves on mud in suspension, which prevents formation of a muddy coastline and meanwhile causes no significant sand transport outside the estuary. Due to the choice for Engelund-Hansen as sediment transport formulation, sand transport processes by waves are excluded.

10     The initial bed composition in the entire domain is $100 \ \%$ sand. In some scenarios mud was supplied to the estuary at the discharge boundary and/or the sea level boundaries. Mud was supplied as a constant concentration, which means that the mass of mud transported into the model depends on the hydrodynamic conditions. For sand supply we used equilibrium conditions at the boundaries, meaning that the capacity of the flow to transport sand at the boundary determined the sand supply rate, which prevents erosion and deposition at the boundaries.

## 2.3   Scenarios

The model was run for 23 scenarios of boundary conditions on the same initial condition. One run constituted 5 hydrodynamics years and 2000 morphological years and took about $20 \ \mathrm{d}$ in real time on one desktop core. The runs with waves took much longer and are therefore quit earlier at $1250 \ yr$.





**Table 3.** Overview of all model scenarios and runs to examine sensitivity to mud-related parameters.

| lab nr | marine mud | fluvial mud | tidal amplitude | discharge | $Pm_{crit}$ | act lyr thickness | settling velocity | waves | |
|---|---|---|---|---|---|---|---|---|---|
| – | $kgm^{-3}$ | $kgm^{-3}$ | m | $m^3s^{-1}$ | – | m | $ms^{-1}$ | – | |
| 001 | 0 | $2e-2$ | 1.5 | 100 | 0.4 | 0.1 | $2.5 \times 10^{-4}$ | no | default, fluvial mud input |
| 009 | 0 | $5e-2$ | 1.5 | 100 | 0.4 | 0.1 | $2.5 \times 10^{-4}$ | no | larger mud input concentration |
| 010 | 0 | $5e-3$ | 1.5 | 100 | 0.4 | 0.1 | $2.5 \times 10^{-4}$ | no | smaller mud input concentration |
| 003 | 0 | 0 | 1.5 | 100 | 0.4 | 0.1 | $2.5 \times 10^{-4}$ | no | no mud, only sand |
| 002 | $2e-2$ | 0 | 1.5 | 100 | 0.4 | 0.1 | $2.5 \times 10^{-4}$ | no | marine mud input |
| 004 | $2e-2$ | $2e-2$ | 1.5 | 100 | 0.4 | 0.1 | $2.5 \times 10^{-4}$ | no | fluvial and marine mud |
| 022 | 0 | $2e-2$ | 1.5 | 0 | 0.4 | 0.1 | $2.5 \times 10^{-4}$ | no | no discahrge |
| 007 | 0 | $2e-2$ | 1.5 | 50 | 0.4 | 0.1 | $2.5 \times 10^{-4}$ | no | smaller discharge |
| 008 | 0 | $2e-2$ | 1.5 | 150 | 0.4 | 0.1 | $2.5 \times 10^{-4}$ | no | larger discharge |
| 021 | 0 | $2e-2$ | 0 | 100 | 0.4 | 0.1 | $2.5 \times 10^{-4}$ | no | no tide |
| 020 | 0 | $2e-2$ | 0.5 | 100 | 0.4 | 0.1 | $2.5 \times 10^{-4}$ | no | much smaller tide |
| 005 | 0 | $2e-2$ | 1 | 100 | 0.4 | 0.1 | $2.5 \times 10^{-4}$ | no | smaller tide |
| 006 | 0 | $2e-2$ | 2 | 100 | 0.4 | 0.1 | $2.5 \times 10^{-4}$ | no | larger tide |
| 029 | 0 | $2e-2$ | 1.5 | 100 | 0.4 | 0.1 | $2.5 \times 10^{-4}$ | yes | fluvial mud + waves |
| 027 | $2e-2$ | 0 | 1.5 | 100 | 0.4 | 0.1 | $2.5 \times 10^{-4}$ | yes | marine mud + waves |
| 028 | 0 | 0 | 1.5 | 100 | 0.4 | 0.1 | $2.5 \times 10^{-4}$ | yes | no mud + waves |
| 025 | $2e-2$ | $2e-2$ | 1.5 | 100 | 0.4 | 0.1 | $2.5 \times 10^{-4}$ | yes | fluvial and marine mud + waves |
| 011 | 0 | $2e-2$ | 1.5 | 100 | 0.2 | 0.1 | $2.5 \times 10^{-4}$ | no | smaller critical mud fraction for cohesive behaviour |
| 012 | 0 | $2e-2$ | 1.5 | 100 | 0.6 | 0.1 | $2.5 \times 10^{-4}$ | no | larger critical mud fraction for cohesive behaviour |
| 013 | 0 | $2e-2$ | 1.5 | 100 | 0.4 | 0.05 | $2.5 \times 10^{-4}$ | no | smaller active layer thickness |
| 014 | 0 | $2e-2$ | 1.5 | 100 | 0.4 | 0.2 | $2.5 \times 10^{-4}$ | no | larger active layer thickness |
| 015 | 0 | $2e-2$ | 1.5 | 100 | 0.4 | 0.1 | $2.5 \times 10^{-3}$ | no | smaller settling velocity for mud |
| 016 | 0 | $2e-2$ | 1.5 | 100 | 0.4 | 0.1 | $2.5 \times 10^{-5}$ | no | larger settling velocity for mud |

We varied fluvial mud input concentration to assess the primary effect on the shape and size of estuaries. Effects of the source of mud was tested by comparing scenarios with fluvial input, marine input, both marine and fluvial input at the same time or no mud input. We further examined effects of waves, river discharge and tidal range. To assess the sensitivity of the model we varied uncertain numerical and process-related parameters: critical mud content for cohesive behaviour, active layer thickness

and settling velocity (Table 3). The model scenarios were analysed by studying the bathymetric changes, mud deposits and geometry of the final bathymetry. These results are compared to each other. In the discussion we compare model results to data of natural estuaries presented above.

About 100 pilot models led us to select the model settings and boundary conditions presented in this paper. For example, we evaluated different initial bathymetries to test its effect on time to equilibrium. We found that the model could both erode

and fill the initial basins for otherwise equal conditions, meaning that the initial shape is only of limited influence on final equilibrium. Moreover, we found that an exponential shape close to the equilibrium size saves considerable computation time and reduces the size of the ebb delta, which then, in turn, has a smaller effect on the incoming tide. Pilot runs with alternative sediment transport formulations confirmed findings of van der Wegen et al. (2008) and led to the choice for the Engelund-Hansen transport equation and a transverse bedslope parameter of 0.2 (Table 2). Furthermore, pilot runs showed that initial

random bed perturbation was unnecessary to trigger bar development. Finally, to test the assumption that the estuaries are well-mixed, the default run was restarted in 3D after 1200 years with salinity and 5 sigma-layers. These results indicated well-mixed conditions, some influence on sediment transport but limited influence on large-scale morphology.



## 3   Results

Here we first describe the general development towards equilibrium of the default scenario with fluvial-fed mud flats. Secondly, we study the hydrodynamics and sediment transport in more detail for the equilibrium condition of the default scenario. Then we describe and compare trends in all scenarios, focussing on mud supply, mud source, effects of waves, river discharge and tidal amplitude. Finally the mud parameter sensitivity runs are presented. Figures with detailed results for scenarios with hydrodynamic variables are shown in the appendix / online supplement.

### 3.1   General development

The final morphology of the default run after 2000 yr with a fluvial mud-supply concentration $of$ 20 $\mathrm{mgL}^{-1}$ is a self-confining bar-built estuary flanked by mud flats and with migrating channels and bars (Fig. 3 d and i). The width of the final morphology decreases exponentially in upstream direction, similar to the initial condition but with self-formed, freely erodible banks.

During the first stage of the development the mud enters the system by river discharge, which is rapidly distributed over the whole estuary within the first few years. The upstream part narrows immediately while narrowing at the mouth starts after about 150 yr and continues for roughly 700 yr (Fig. 3 b and c). After 200 yr the sand within the estuary is redistributed and the ebb delta starts to form. The ebb delta continues to prograde as sand and mud are supplied constantly whilst coastal sediment transport is absent. Since we are not interested in the evolution of the ebb tidal delta and littoral processes are not well modelled in this setup, the area downstream of the coastline is excluded in further analyses.

Within the first 3 yr the upstream part of the estuary starts meandering and the downstream part starts braiding. Meanders grow and migrate downstream while bifurcations develop and chute cutoffs occur. Within 200 yr, the an initial bar pattern has developed throughout the estuary, and the channel pattern is characterised by mutually evasive, ebb or flood-dominated channels (Fig. 3 b and c). The bars continue to migrate downstream throughout the simulation as an effect of the fluvial discharge and sediment input. After about 1000 yr the bar-channel pattern appears to have reached a dynamic equilibrium with channels of approximately 4 m deep and bars elevated to mean water level.

Morphodynamic equilibrium, where average bank erosion equals sedimentation, is indicated by a net bed-level change fluctuation around zero (Fig. 3j). This means that there is no net accumulation or erosion in the estuary, so no net import or export of sediment. Furthermore, we observe that the absolute bed level changes approach a constant value (Fig. 3e). The initial changes in which the estuary adapts to the boundary conditions (like width and depth adaptation) happen within centuries, while the dynamic behaviour of bars and channels continues throughout the simulation, as also shown by the constant nonzero value approached in Fig. 3e. If the mean of the absolute bed level changes approached zero then the bathymetry would have become fixed. It could be argued that the lowering in Fig. 3e indicates that a true equilibrium was not reached and will not be reached because the river continues to import sand and the ebb delta continues to grow in the near-absence of littoral processes, but for our purposes and time scale of interest Fig. 3j and the timeseries of maps indicate that equilibrium planform geometry of channel, bars and mudflats in the estuary was reached after about $500 - 1000$ yr.





**Figure 3.** Results of the default run with a fluvial mud supply of $20\,\mathrm{mgL}^{-1}$, 3 m tidal range and $150\,\mathrm{m}^3\mathrm{s}^{-1}$ river discharge. (a-d) Bathymetry after $50, 150$ and $2000\,\mathrm{yr}$, (f-i) mud fraction in the top layer of the bed after $50, 150$ and $2000\,\mathrm{yr}$, (e) morphodynamic activity expressed by absolute bed level change over time between the original coastline and the upstream boundary, (j) net bed level change over time between the coastline and the upstream boundary, positive is net accretion and negative erosion. Movie of the model is available on youtube: $https:$
$//youtu.be/HAeka4e2_PY$





## 3.2 Hydrodynamics and sediment transport

Tidal water levels and velocity vary along the estuary: at the seaward boundary the tide is a symmetrical M2, while further into the estuary the tide becomes asymmetrical (Fig. 4). At the mouth the water level rapidly progresses from low to high water and slowly progresses from high to low water. There is no phase lag between water level and velocity. The tidal range decreases

further into the estuary mainly by a decrease in the low water amplitude (Fig. 4a). Likewise, flood velocities reduce while ebb velocities remain about constant (Fig. 4b, e and f). Additionally the duration of the ebb flow is longer (Fig. 4b and g). The ebb flow increases in relative velocity and duration further upstream, because the contribution of the river increases in this direction. The tidal excursion length is a little over $6 \, \mathrm{km}$. The location of the 1ppt isohaline is $5 \, \mathrm{km}$ upstream of the estuary mouth during high tide, which was inferred from the 3D restart of the default scenario with salinity. In the Dovey estuary the

1ppt isohaline is around $7.5 \, \mathrm{km}$ and also well mixed (Baas et al., 2008). We consider this is in good agreement because the model discharge is larger and the spit is ignored.

These hydrodynamics might suggest that the estuary is still an exporting system and not in equilibrium. However, spatial variation is very important. We observed flood dominant velocity amplitudes over shallow areas like bars and mudflats and ebb-dominant velocity amplitudes in the channels (Fig. 4h) so that flood discharge and ebb discharge balance but for the net river

inflow. This is in agreement with most research about tidal asymmetry (e.g. Speer and Aubrey, 1985; Friedrichs and Aubrey, 1988; Wang et al., 2002; Robins and Davies, 2010; Brown and Davies, 2010). This research strengthens these findings, because this estuary is self-formed, while several bathymetries tested in previous research are strongly simplified or arbitrary chosen and might not represent a realistic state of an estuary (Speer and Aubrey, 1985; Friedrichs and Aubrey, 1988; Robins and Davies, 2010; Brown and Davies, 2010).

SPM levels reach the highest local concentrations of $45 \, \mathrm{mg/l}$ between 2 and $4 \, \mathrm{h}$ after high tide with a typical mean concentration similar to the input concentration of $20 \, \mathrm{mgL^{-1}}$. The mean SPM levels of the Dovey are comparable and estimated to be $32 \, \mathrm{mgL^{-1}}$ (Painting et al., 2007). The typical non-dimensional shear stress (Shields number) of the model is 0.27. Over the whole model run $4000 \, \mathrm{m^3}$ of sediment is imported into the estuary of which $7800 \, \mathrm{m^3}$ of mud is imported and $3800 \, \mathrm{m^3}$ of sand is exported. Cleveringa and Dam (2013) found similar trends in a separated analysis of the sand and mud balance in the

Western Scheldt.

In the final stage of the model, net sediment transport is in the ebb direction for bedload transport and suspended transport, i.e. of sand and mud (Fig. 4d). Notably the amount that is transported through the mouth (solid line) is equal to the sediment input from the river (dotted line). This shows that there is no net deposition or erosion in the estuary in agreement with Fig. 3d, meaning that the estuary is in equilibrium.

The river discharge is about $7.5 \, \%$ of the maximum tidal flood discharge and contributes to the ebb flow. Therefore the volume of water flowing through the mouth during ebb is always slightly higher than in flood direction, although the difference is not very large (Fig. 4c). Since the duration and the velocity amplitude asymmetry are ebb dominant, cross-sectional flow area is larger for the flood flow otherwise there would be an imbalance in tidal prism: $Q = u \times w \times h$; $P = Q \times t$.





**Figure 4.** Hydrodynamics of the last day after 2000 yr. Left panels show temporal variation in one day and right panels show spatial variation along estuary. (a) Water level, (b) streamwise flow velocity in the deepest channel with negative velocity towards the sea, (c) instantaneous tidal discharge through cross section and (d) cumulative sediment transport through the cross-section showing no net difference between the upstream boundary and the coastline. (e) Maximum peak velocity for ebb and flood, (f) ebb and flood duration, (g) peak ebb and flood velocity ratio and (h) spatial pattern of peak velocity ratio showing flood dominated shallow areas. Solid lines in e and g are based on streamwise velocity and dotted line is based on velocity magnitude, showing effects of bends at 10 km.



### 3.3 Effects of mud flat formation

We will now compare other scenarios with the default run. In most scenarios mud is accumulating on the flanks of the estuary where the velocities are low and in the upper estuary where it covers a relatively large fraction of the width (Fig. 3b). Locally, mud accretes on bars that are rather stable (e.g. Fig. 3b, on the ebb delta). The initiation of mud flats proceeds by the positive

feedback identified in the model description: once mud starts settling somewhere, the mud fraction in the bed rapidly increases beyond the critical mud fraction for mud- dominated behaviour. As a consequence, the critical shear stress for sand erosion increases, thus becomes more difficult to erode and more rapid aggradation of mud is likely to occur.

Migration rates of channels decrease considerably due to the addition of cohesive material (Fig. 8a–h). Bar splitting and merging related to chute cut-offs and avulsion also reduce with increasing mud concentrations. In Fig. 8–h channels move

through a cross-section at the mouth through time, though slower for a larger mud supply. For example, a large bar forms in the mouth after about $1100\,\mathrm{yr}$ in the scenario with only sand (Fig. 8a) and in the scenario with a mud supply of $50\,\mathrm{mgL^{-1}}$ (Fig. 8d). In the run with mud, the bar is covered with mud and becomes fixed while the large bar in the scenario with only sand migrates about $1\,\mathrm{km}$. Absolute bed level changes also indicate that dynamics are decreased with mud input (Fig. A3y-II), because there is less bed level change per timestep.

The mudflats have a strong effect on the final shape of the estuary (Fig. 5). Firstly, an increase in fluvial mud input concentration leads to stronger self-confinement of the estuary. By depositing mud on the sides of the estuary, the banks become more stable and limit (further) erosion due to an increased critical shear stress. Self-confinement of estuaries is clearly observed when the models with mud supply are compared to the control run without mud (Fig. 5a). The runs with mud are narrower and have a smaller surface area due to filling of the initial bathymetry, while the sand run has expanded in size. Consequently, the

braiding index lowers with increasing mud concentration (Fig. 7e-h). In contrast, estuarine surface area continues to increase over time for the control run with only sand (Fig. 8q). After the initial rapid change the increase in area and width is linear, driven by dynamic channels and bars and is unhindered by bank stability. This suggests that there is no equilibrium shape under these conditions as is also reflected in the absolute and net bed level change (Fig. 8y-VI). The absolute bed level change does not approach a constant value and the net bed level remains negative, demonstrating the sand-only estuary to be a continuously

exporting system.

For estuaries with fluvial mud, higher concentrations lead to narrower (Fig. 7i–l and Fig. 8i–l) and smaller (Fig. 8q–t and Fig 6) estuaries. Moreover, in some places the width of the estuaries with mud supply is narrower than the initial width, supporting our finding that the initial bathymetry is of limited influence because the system is able to fill and to expand (see methods). Furthermore, tidal bars become higher with increasing mud concentrations, which results in an increased average

bed level (Fig 7a–d). Furthermore, mud is almost nowhere deposited in the channels and does therefore not limit bed erosion by cohesion (Fig. 6). We therefore infer that the shallower channels in increasingly muddy estuaries mainly result from the decrease in estuary width and concurrent reduction of intertidal area, tidal range and tidal currents (Fig 7).

With larger mud concentrations a larger area of the estuary is covered with mudflats (Fig. 7m-p). The mud cover maps (Fig. 5e-h) indicate that although the distribution of the mud is quite similar for different concentrations, the overall mud cover



**Figure 5.** Effects of mud supply concentration. Left column shows final bathymetry of model runs after 2000 yr and the right column shows mud fractions in the top layer of the bed. Runs with (a,e) 0 mgL$^{-1}$, (b,f) 5 mgL$^{-1}$, (c,g) 20 mgL$^{-1}$ (default) and (d,h) 50 mgL$^{-1}$ fluvial mud supply concentration.

over the estuary length increases with mud input concentration (Fig. 7m-p). In general, more mud leads to wider mud flats on the side and seems more likely to deposit mud on mid channel bars. The maximum fraction of intertidal area shifts from the middle estuary to the lower estuary for increasing mud concentration. At the same time mud increasingly deposits on lower elevations (Fig. 6) as seen in the strong increase in cumulative area just above mean water level.

5    The estuaries with mud are shorter than the estuary with only sand. The length of the estuary is defined as the distance between the mouth and the limit of tidal influence (where tidal range reaches zero in Fig. 7q–t). Estuaries are shorter for scenarios





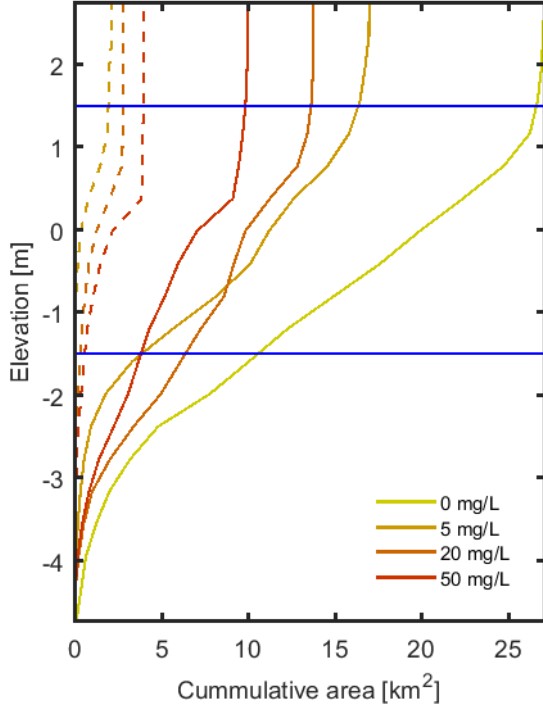

**Figure 6.** Hypsometric curves of the final bathymetry after 2000 yr. Curves indicate cumulative area below a certain elevation. Dotted lines indicate the mud covered area at below this elevation. Runs with $0,\ 5,\ 20$ and $50\ \mathrm{mgL}^{-1}$ fluvial mud supply concentration.

with mud compared to the scenario with only sand, but mud concentration seems irrelevant. Mud supply concentration has some effect on the funnelling of the estuary, although the temporal variation of the funnel-shape is less for higher concentrations (Fig. 8m–p). In general, funnel-shape strength first decreases and then increases again. This has to do with the order of widening and narrowing of different parts of the estuary. The width of the mouth always decreases at the start of the run, after about $400$ yr that can change into widening or continue (Fig. 8i–l). This narrowing at the start increases the funnel-parameter. Upstream, the estuary width initially increases which also decreases funnelling. The e-folding length of the scenario without mud supply is not the shortest compared to the other scenarios, but the run without mud results in a larger estuary. For a longer convergence length, the funnelling-parameter can be the same if the estuarine mouth is bigger.





**Figure 7.** Hydrodynamics and morphology along estuaries with different mud supply concentrations. From left to right column: model with only sand, mud supply concentration of 5, 20 (default) and 50 $mgL^{-1}$. (a–d) Minimum, mean and maximum bed elevation and high and low water level, (e–h) braiding index, (i–l) estuary width defined as: the initial width, maximum reach over the whole scenario run, the width of wet cells in the model, width defined by a threshold value that is used to mask the cells that are around the dry-wet cell threshold. (m–p) intertidal area and mud cover as percentage of the total area, (q–t) tidal range and (u–x) peak ebb and flood velocities.




**Figure 8.** Hydrodynamics and morphodynamics over time for estuaries with different mud supply concentrations. From left to right column model with only sand, mud supply concentration of 5, 20 (default) and $50 \, \mathrm{mgL^{-1}}$. (a–d) Bathymetry of the cross-section at the mouth plotted over time, (e–h) mud fraction in top layer of cross-section at the mouth, (i–l) estuary width at 1, 4 and 8 km from the mouth, (m–p) funnel-shape parameter, (q–t) estuarine surface area, (u–x) intertidal area and mud in the bed relative to the total area, (y–II) absolute bed level change and (III–VI) net bed level change.



### 3.4 Difference between fluvial and marine mud supply

Estuaries develop very differently when mud is imported from the sea rather than from the river under the assumption that the mud characteristics are the same. For marine mud, the mud flats form only in the lower estuary up to $9 \, \mathrm{km}$ upstream from the mouth, because mud can only occur in regions where there is significant flood flow to transport the mud upstream. For fluvial mud on the other hand, mudflats form along the entire length of the estuary. Mud supply from both boundaries simply has a combined effect with mud distributed along the whole estuary and the highest mud cover near the mouth.

Estuaries are narrower with fluvial mud supply compared to the marine mud supply and the sand-only control run. In case of marine mud supply the estuary decreases in width near the mouth, but is upstream similar in width and bed level to the estuary without cohesive sediment. In the first $500 \, \mathrm{yr}$ the width at $1 \, \mathrm{km}$ from the mouth decreases and is partly taken in by mud flats, but returns to the initial width after $2000 \, \mathrm{yr}$. On the other hand, estuary width increases at $4$ and $8 \, \mathrm{km}$ from the mouth. For the scenario with both mud from the sea and the river, the estuary mouth is narrower than for only marine or fluvial mud.

The total estuary area continues to increases for the scenario with marine mud supply, because the upper estuary widens similar to the run without mud. Likewise, the estuary is not confining itself by cohesion and does therefore not reach equilibrium. The estuaries that include fluvial mud supply eventually reach a constant area over time and do reach equilibrium.

The estuary with fluvial mud supply shows strong funnelling due to more narrowing between $5 - 10 \, \mathrm{km}$ than between $0 - 5 \, \mathrm{km}$ from the mouth. The estuary with marine mud-supply shows an opposite trend with stronger narrowing near the mouth, leading to a lower convergence. The length of the estuary is shorter for scenarios that include fluvial mud supply and the tidal range and flood velocity along the estuary decrease faster.

Not all mud settles in the estuary, but a lot is transported out of the estuary or never enters the estuary from the seaward boundary. Part of this mud is deposited at the coastline and part is transported out of the model domain. Because mud is supplied as a concentration depending on the discharge, a much larger volume of mud is supplied to the system when the mud is supplied by the sea. This large volume of mud causes significant deposition at the coast and affects morphology at the mouth. We consider this an artefact due to the lack of littoral processes.

### 3.5 Effects of hydrological boundary conditions: river, tide and waves

Changes in the boundary conditions in the form of tidal amplitude and discharge did not seem to alter the location of the mudflats, but only the size. More and larger mudflats formed with higher discharges. An optimum in mud flat size occurred for increasing tidal amplitude. With lower amplitudes there is less intertidal area and therefore less space for mudflats, and with higher amplitudes the higher velocities prevent deposition. There is a balance between the tidal flow and fluvial flow into the estuary. When the river becomes more important tidal damping occurs under the influence of increased river discharge by friction (Horrevoets et al., 2004), therefore the point of tidal influence is further downstream decreasing the tidal prism and therefore tidal velocity. This means that the excess width can be filled until the appropriate width-depth ratio of the river to this point. When the river has less influence, the tidal intrusion is larger with higher velocities. This balance influences the





morphology: relative stronger tidal influences lead to larger estuaries when the transition to more river dominated estuaries decrease in size, fill up and eventually evolve into deltas.

More specifically, no river discharge leads to large tidal meandering channels in the lower estuary with a filled upper estuary. On the other hand, larger discharges lead to a transition from filled estuaries to a delta. Besides a change in the river-tide

balance, increased discharge means more sediment input at the equilibrium boundary condition used for sand. Additionally, mud is supplied as a concentration and mud volumes therefore increase with higher discharges. As a result the system rapidly expands the ebb tidal delta, fills the estuary and transforms into a delta for the highest discharges. This means that the balance between fluvial discharge and sediment supply and the tide and tidal sediment export is changed.

The estuary shape scales with discharge, but size does not. Lower discharge leads to stronger funnelling of the estuary.

On the other hand, size hardly changes with discharge despite the fact that larger discharges result in more vigorous channel migration and faster dynamics. We only observe a sudden transition in size from estuary to delta between a discharge of 100 and 150 $\mathrm{m^3 s^{-1}}$. Adversely, tidal amplitude has a strong effect on the size of the estuary. In fact, a tidal amplitude of less than 2 m leads to closure of the estuary and formation of a muddy delta. The larger flow velocities with higher tidal range keep mud in suspension so that less mud settles in the estuary, in turn leading to less self-confinement. Systems with lower tidal

amplitudes are therefore more likely to develop deltas rather than equilibrium estuaries. Further it is observed that larger tidal ranges lead to larger tidal meanders and bigger channels. An additional effect is that higher velocities due to increased tidal amplitude cause enhanced shifting of the channels, which prevents the settling of mud on the bars sufficiently to change the erosional behaviour and prevent the positive feedback of mud to kick in. This effect is also caused by waves.

Waves prevent mud deposition at the coastline, instabilities in the sea area and cause widening of the mouth. This especially

leads to limited influence of marine mud supply, though it is supplied 5 km further upstream with waves. For example, the run with marine mud supply and waves is very similar to the run without mud supply with waves. The addition of waves also result in higher high water levels and therefore a larger tidal range and higher flood velocities near the mouth, especially flood velocities. These higher velocities contribute to widening at the mouth leading to a very strong funnel shape. Due to the waves there is little mud cover in general, only upstream.

## 3.6 Effects of sediment transport parameters

The sensitivity to active layer thickness was assessed by doubling and halving the active layer did not lead to different large-scale trends in mud flat formation and estuary shape and dimensions. A different active layer thickness leads to a different pattern, but the large-scale characteristics of the pattern are the same. Likewise, the critical mud fraction that determines cohesive and non-cohesive behaviour had no significant effect on large-scale morphology. Initially there is slightly more dynamics

in the run with the higher critical mud fraction, but this effect can be disregarded after some time. On the other hand, the order of magnitude of the settling velocity had a considerable effect: a 10 times slower settling velocity resulted into an estuary with more similar geometry to the run with only sand while 10 times higher settling velocities developed a delta due to larger sedimentation rates. This means that similar trends would probably be found for increasing mud fraction with lower settling velocity for higher concentrations and addition of biotic effects on apparent cohesion, and similar trends would probably be





found for higher settling velocities with larger tidal amplitude. We predict that general trends and conclusions will not be affected by changes in mud characteristics like settling velocity, erosion parameter and critical shear stress for erosion, but might lead to slightly different equilibria.

5  We did not test the combined effect of changing the proportions of clay and silt, whereby the settling velocity and critical shear stress for erosion would probably be inversely correlated and have opposite effects, reducing the effects of these parameters. Additionally, we ignore consolidation which especially affects layer thickness and erosion characteristics of mud layers. With this in mind we expect that the migration of deep channels that are eroding deep, old mud layers is overestimated. Additionally, we assume that the time in which thick mudflats are developed is also overestimated and the critical shear stress of very recently deposited mud in reality is also overestimated due to fluff characteristics of mud when it is still submerged.

Mostly we expect effects timewise, but not in the general pattern and trends.

## 4   Discussion

The most important findings from the results are summarised in Fig. 9. Mud supply leads to self-confinement of the estuary by the development of mudflats on the sides (Fig. 9d). We observed that larger mud supply concentrations leads to narrowing and filling of the estuary towards a dynamic equilibrium, while the estuary without mud supply continued to widen and grow

in size (Fig. 9d,g). Furthermore, we observe that mud raises the bed level, decreases the length, increases mud flat size, decreases dynamics and increases funnelling (Fig. 9a). Marine mud supply causes the development of a muddy coast and in this model only influences the mouth of the estuary, which strongly decreases funnelling of the estuary, but is of little influence in combination with waves though these effect might be underestimated due to uncertainties in wave transport and chosen settling velocities. In scenarios with larger fluvial mud supply, larger flow discharge plus fluvial mud supply, and lower tidal

amplitude the initial estuary shape was filled and a delta developed (Fig. 9). By this we mean that the deltaic channels had only negligible tidal flow and were much smaller than the initial estuary. These results suggest a rather sharp transition from a narrow equilibrium estuary with significant tidal action to an extending river-dominated delta.

### 4.1   Comparison to real estuaries

Model conditions fall within the parameter space of natural estuaries (Fig. 11; Table 4; Prandle et al., 2005; Leuven et al.,

2016). The model has typically larger discharges than the small UK estuaries, but discharge and tidal amplitude falls well within the range of estuaries worldwide.

 Several aspects of the bar patterns are further indications that the numerical models reproduce important emergent phenomena of real estuaries. For example, we observe ebb- and flood-dominated channels that are unique for tidal systems (van Veen, 1950; Ahnert, 1960). Typical bar dimensions obtained from the models are in good agreement with natural estuaries from a

large dataset (Leuven et al., 2016); for example tidal bar length is approximately 7 times the partitioned bar width (maximum bar width devided by barb channels). Furthermore, bar length approximates local width of the estuary. Bars without mud are generally longer and wider for this model study and relative to the local estuary width. Bars in models are also slightly bigger





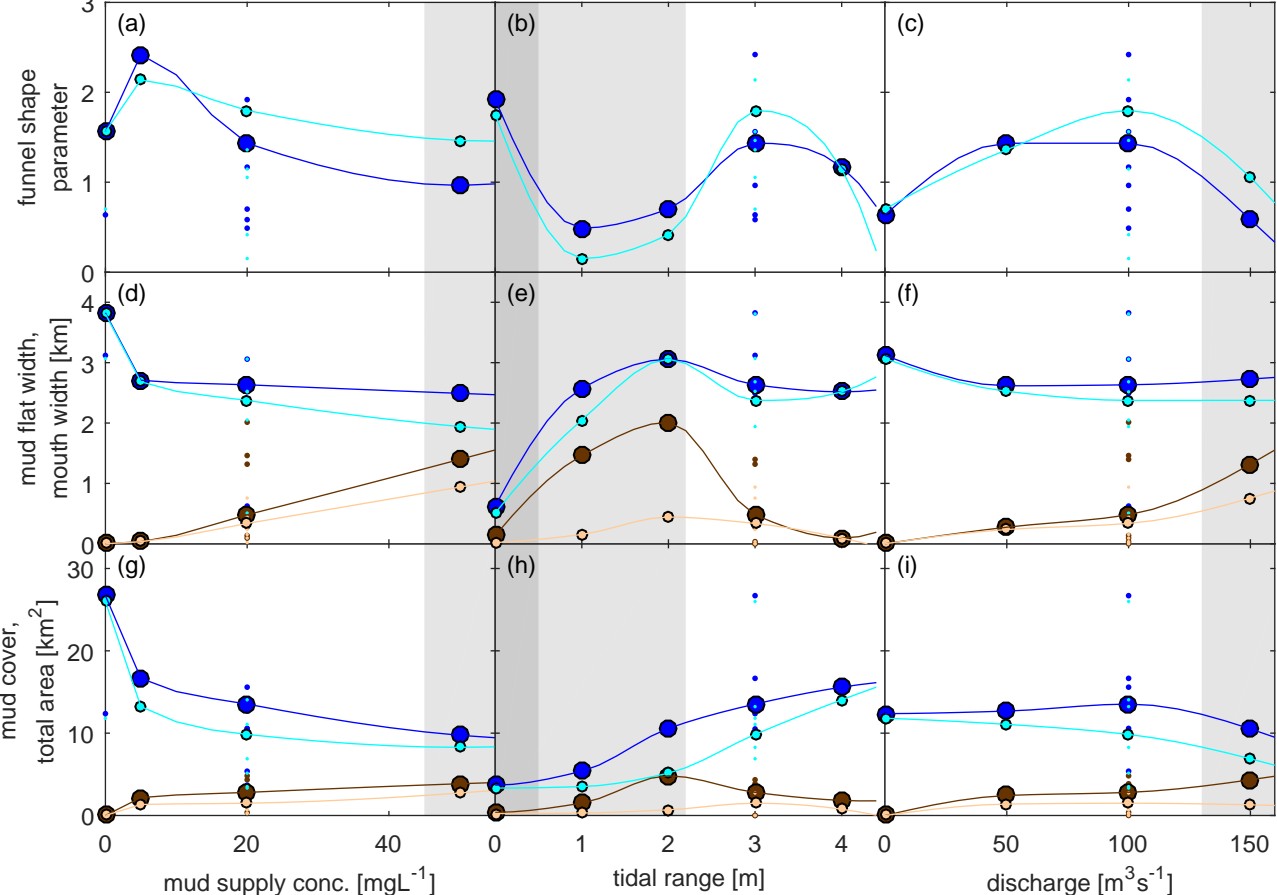

**Figure 9.** Most important large-scale morphological parameters as a function of the varied boundary conditions: fluvial mud supply concentration ($\mathrm{mgL}^{-1}$), tidal range (m) and fluvial discharge ($\mathrm{m}^3\mathrm{s}^{-1}$). (a-c) funnel-shape parameter, (d-f) mouth with (in blue colours) and mud flat width (brown colours) at the mouth (km) and (g-i) total area (blue colours) and mud covered area (brown colours; $\mathrm{km}^2$). Data indicated in light blue and light brown use a more aggressive masking technique in which high mud flats are masked from the estuary shape from which area, width and funnel-shape are calculated. Light grey areas indicate models in the transition from estuary to delta. Dark grey indicate models that evolved into a delta.

with marine mud supply rather than for fluvial mud supply. The braiding index is strongly related to estuary width as found for natural estuaries (Leuven et al., 2016) and in agreement with the relation between tendencies to form floodplain in rivers and the resulting relation between channel aspect ratio and bar pattern (Kleinhans, 2010; Kleinhans and van den Berg, 2011; Schuurman et al., 2016).

5    The completed model runs show mud flat characteristics and behaviour broadly comparable to natural estuaries. Trends in the field data, shown earlier, generally agree well with the model results. Comparison of the observed and modelled hypsometries (Fig. 6 and Fig. 1g and h) shows that mud is deposited at comparable elevations, namely around mean water level. We observe





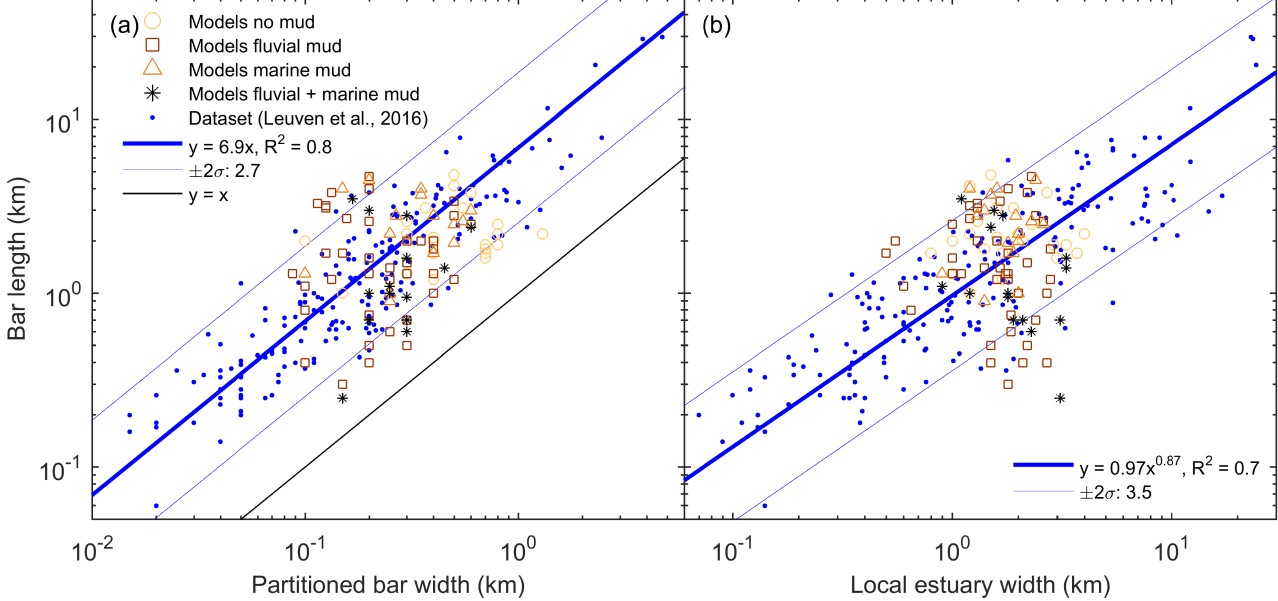

**Figure 10.** (a) Bar length versus partitioned bar width and (b) bar length against local estuary width. Model results plot in the same range as the data of the natural estuaries (Leuven et al., 2016).

similar depositional areas of mud on the sides of the estuaries in the form of mudflats (Fig. 1a–e and Fig. 5e–h). In the centre of the lower estuary there is little mud compared to the mudflats on the sides. However, some mud is observed on some of the bars in the Western Scheldt (e.g. Fig. 1c) as in some model scenarios (Fig. 5h).

The fluvial mud scenarios have relatively large fractions of width covered by mud flats in the upper estuary as in the single-channel upper estuaries in the Netherlands. Indeed, most mud is deposited in the middle and upper estuary, where the estuary consists of only one channel. This is also clearly observed in the McLaren dataset of the Western Scheldt (Fig. 1a). The tidal river was found to contain more mudflats than the lower estuary (Fig. 1f). Note that Fig. 7 underestimates modelled mud flat surface shown in Fig. 5 because many cells are inactive in the computation because they increased in elevation.

Typically in the model, marine mud does not settle much and far in the estuary. This is not what is observed in the Western Scheldt. Verlaan (2000) studied the marine versus fluvial distribution of mud through the estuary. He found a sharp increase in mud fraction in the bed between Lillo and Saeftinge from 10 % to 75 %, which is far upstream in the narrow single channel system. This might be a consequence of the assumption that settling velocities for fluvial and marine mud are the same while typically settling velocities of marine mud are significantly higher. It is also a likely possibility that the Western Scheldt is not comparable to our modelled system considering marine mud deposits, because the salinity intrusion of the Dovey and Western Scheldt is incomparable. Mud deposition data from the Dovey estuary is unavailable although mud flats and muddy marshes are easily observable on aerial imagery (Leuven et al., in review).




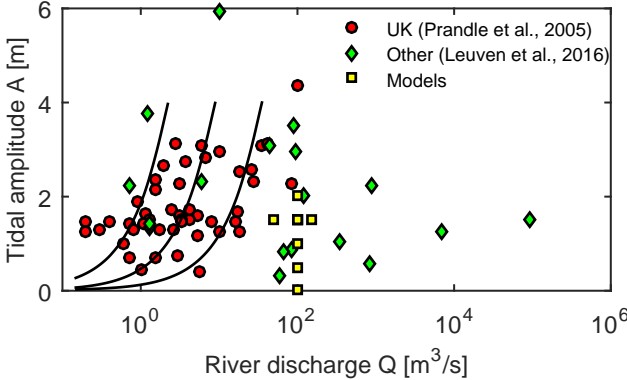

**Figure 11.** Tidal amplitude plotted against river discharge for real world estuaries and modelled scenarios. Field data is used from Prandle et al. (2005) for estuaries in the UK and several other sources for different estuaries over the world. Lines indicate estimations of estuarine length by Prandle et al. (2005) of $5, 10$ and $20$ km

In the model we observe sharp transitions between areas without mud in the bed ($< 10$ %) and areas with very high mud fractions ($70 - 100$ %). This is also observed in the Western Scheldt according to van Ledden (2002). More gradual transitions of mud are expected for $w_s \times c/M >> 1$, where $w_s$ is fall velocity, $c$ is concentration and $M$ is the erosion parameter (van Ledden, 2002). All our models have ratios below 1 in agreement with conditions in the Western Scheldt and probably in
agreement with conditions in the Dovey given the clearly observable sand-mud transitions on imagery.

In the Western Scheldt the fluvial mud supply varies between $100$ and $300$ Ggyr$^{-1}$ at the Rupple mouth (Taverniers, 2000). In the model the mud input is $63$ Ggyr$^{-1}$. The mean discharge of the Scheldt, about $120$ m$^3$s$^{-1}$, is about $20$ % higher than the default model scenario, while the sediment input is at least $60$ % higher. This higher mud load might explain why the Western Scheldt has more mud deposits. In the field case this occurs more on bars than on the sides compared to the models, which
difference we may partly attribute to the embankment and limited space to form mud flats and partly attribute to spatially and temporally varying mud characteristics in the Western Scheldt.

### 4.2   Transition from estuary to delta

The parameter space of Prandle et al. (2005) suggests that tides and river flow are sufficient conditions to explain the bathymetry of an estuary, with longer tidal reaches with larger river inflow (Fig. 11). This trend is not reproduced in the idealised model
scenarios that typically have a tidal reach of $5 - 15$ km long, but plot far above the line of $20$ km in Fig. 11. Likewise, the trend is not clear in the dataset either (Fig. 3 in Prandle et al., 2005). Rather, we observe the opposite trend: shorter estuaries, or even deltas, form with larger river discharges and longer estuaries form in higher tidal ranges. Possibly longer estuaries form for larger total flow from the combination of tide and river. We found much stronger effects of mud supply, suggesting that the tide-discharge parameter space needs to be extended with sediment supply.





**Table 4.** Ranges of conditions in mixed estuaries at temperate zones (Prandle et al., 2005) compared to values for the modelling results.

| Parameter | Unit | Range | Model |
|---|---|---|---|
| Tidal amplitude | m | $1-4$ | 1.5 |
| Velocity amplitude | $\mathrm{ms}^{-1}$ | $0.5-1.25$ | $0.5-1$ |
| River discharge | $\mathrm{m^3s}^{-1}$ | $0.25-3000$ | 100 |
| Associated current | $\mathrm{ms}^{-1}$ | $0.001-0.01$ | ?? |
| Flushing time | d | $1-15$ | ?? |
| Depth at the mouth | m | $1-20$ | 2 |
| Tidal intrusion length | km | $2.5-100$ | 15 |
| Age | yr | $100-15000$ | 2000 |
| Fall velocity | $\mathrm{mms}^{-1}$ | $0.5-5$ | $0.25(mud), 41(sand)$ |

As our model runs cover transgressive and regressive trends as effects of tides, river, waves and sediment supply on morphology we attempted to position our results in the traditional ternary classification diagrams for deltas of Galloway (1975). An expanded version of this classification system includes all coastal environments, where larger river influence leads to delta development and low or absence or river influence leads to lagoons, strandplains and tidal flats (Dalrymple et al., 1992; Boyd

et al., 1992). Qualitatively our results also show that for higher river discharge the estuarine system transitions to a deltaic system (Fig. 9c–i) by filling of the estuary. Note that the width did not decrease because a small tidal basin north of the river mouth affected the automated calculation of the width of the system (Fig. A4a). We also observed a transition to deltas when the tidal range was decreased (Fig. 9b–h), so that the relative power of the river increases in qualitative agreement with the classification diagram.

However, the most important findings of our research are more difficult to relate to these diagrams. We found that an increase in mud supply concentration leads to confining and filling of the initial estuary shape (Fig. 9a–g) leading to a decrease in total area and width at the mouth, while the mud covered area and mud flat with at the mouth increased and is more delta-like. Orton and Reading (1993) found that smaller grain size leads to narrower channels in deltas and a tendency to avulse rather than have migrating channels. We observe similar behaviour in the model scenarios but here this is related not merely to grain size but to

the supply rate.

Alternatively, Dalrymple et al. (1992) and Boyd et al. (1992) developed a classification system with a fourth dimension based on the evolution of coastal systems by defining it as a prograding or transgressive system on the basis of sea level rise and sediment supply. This system disregards the possibility of an equilibrium without progradation and without transgression through combinations of sediment supply but otherwise similar hydrodynamic conditions. Our models with different fluvial

mud supply concentrations lead to distinct different morphologies but would plot on the same coordinates in these diagrams. Additionally, sea level rise is an ambiguous and qualitative variable in their conceptual figure, because it affects the hydrodynamic conditions of the primary ternary diagram. To conclude, our model results for estuaries qualitatively fit in the ternary plots





of Dalrymple et al. (1992) and Boyd et al. (1992) for deltas when sea level rise is ignored and sediment supply is considered the only variable on the fourth axis.

## 4.3 Large-scale equilibrium of estuary shape and dimensions

Estuaries with fluvial mud supply evolve to large-scale morphodynamic equilibrium (where absolute bathymetry change is
constant, Fig 3c, net bathymetry change is zero, Fig 3d, and net export equals import, Fig 4d) with dynamic channels and bars, but in the absence of mud keep expanding continuously by bank erosion due to channel migration. This agrees with the continuously exporting estuaries in the numerical models of van der Wegen et al. (2008) and with the physical experiments of Kleinhans et al. (2015) with perpetually expanding tidal basins in cohesionless sand. After a rapid adjustment of basin size and bar and channel pattern the experiments developed to near-equilibrium but never attained equality of sediment import
and export. Our scenario without discharge is similar to these experiments and shows the same evolution, including the rapid adjustment and continuous erosion in a low-dynamic state (Fig. A6d-VI). In braided rivers such unhindered bank erosion leads to a 'threshold channel' (Parker, 1978) with an equilibrium width related to the upstream flow discharge and the threshold for sediment motion. This theory was earlier suggested to be valid for tidal basins (Kleinhans et al., 2015). However, unlike rivers, estuaries are not limited by discharge because tidal prism can continue to increase as the estuary enlarges, leading to a
potentially positive feedback only limited by friction. In other words, estuaries may expand to much larger systems because the tidal prism adapts to estuary size and flow velocities and entrainment rates will not decrease with basin size unless opposed by cohesion. This proved to be the case in our models with mud. From this we conclude that development to an equilibrium shape for estuaries requires some form of apparent cohesion from mud, from species with sediment-binding effects and from unerodible valley walls.

This explains why previous studies found large-scale equilibrium in estuaries: these imposed a fixed estuary shape and size in 1D simulations (e.g., Lanzoni and Seminara, 2002; Schuttelaars and de Swart, 2000; Todeschini et al., 2008) or imposed non-erodible boundaries in 2DH (e.g., Hibma et al., 2003; van der Wegen and Roelvink, 2008).

The novel model applications and results open up possibilities to incorporate effects of species on flow and sediment transport (van Oorschot et al., 2015), where species and species density depend on substrate and salinity, and to unravel effects of initial
conditions inherited from early Holocene systems from effects of boundary conditions (Townend, 2005).

## 5   Conclusions

The size and shape of alluvial river estuaries depend strongly on the supply of mud, because this determines mud flat formation that protects erodible estuarine boundaries against erosion. This was concluded from a series of idealised morphological model runs for medium-sized estuaries with sand and varying concentrations of mud, a range of tidal amplitudes and river discharge,
and limited littoral processes. Estuaries with mud supply may develop a dynamic morphological equilibrium, on the other hand, estuaries with only sand in the bed and banks expand perpetually with a positive feedback between tidal prism and sediment export. This means that freely developing estuaries self-confine their size and reduce channel and bar dynamics with increasing





fluvial mud supply. Within centuries they attain a large-scale equilibrium with balanced sediment import and export. Higher mud supply concentrations result in shorter, shallower, narrower and in general smaller estuaries with increasing mud flat area and stronger funnelling, that may develop into tidal deltas depending on the littoral conditions. Spatial patterns of mud flat development in estuaries depend strongly on whether the mud originates from the sea or the river: marine mud only influences

5 the lower estuary with these model conditions, while fluvial mud deposits along the entire system in qualitative agreement with field data. The effect of marine mud supply is even less when waves are included, even though mud is transported further upstream. Tidal range and river discharge have opposing effects on the balance between mud deposition and erosion. For higher fluvial mud concentrations, relatively high river discharges and low tidal amplitudes estuaries transition into prograding deltas. These general trends are similar to effects of floodplain formation and erosion on the width and bar pattern in rivers.

10 **6   Data availability**

Delft3D input files of the default model will be added as supplementary material.





**Figure A1.** Effects of mud source. Left column shows final bathymetry of model runs after 2000 yr and the right column shows mud fractions in the top layer of the bed. Run with (a,e) only sand, (b,f) marine mud input (default), (c,g) marine and fluvial mud input and (d,h) fluvial mud input.

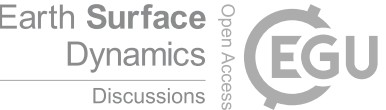

**Figure A2.** Hydrodynamics and morphology along estuaries with different mud sources. From left to right column: model with only sand, marine mud supply, supply from both boundaries and fluvial supply (default). (a–d) Minimum, mean and maximum bed elevation and high and low water level, (e–h) braiding index, (i–l) estuary width defined as: the initial width, maximum reach over the whole scenario run, the width of wet cells in the model, width defined by a threshold value that is used to mask the cells that are around the dry-wet cell threshold. (m–p) intertidal area and mud cover as percentage of the total area, (q–t) tidal range and (u–x) peak ebb and flood velocities.



**Figure A3.** Hydrodynamics and morphodynamics over time for estuaries with different mud sources. From left to right column: model with only sand, marine mud supply, supply from both boundaries and fluvial supply (default). (a–d) Bathymetry of the cross-section at the mouth plotted over time, (e–h) mud fraction in top layer of cross-section at the mouth, (i–l) estuary width at 1, 4 and 8 km from the mouth, (m–p) funnel-shape parameter, (q–t) estuarine surface area, (u–x) intertidal area and mud in the bed relative to the total area, (y–II) absolute bed level change and (III–VI) net bed level change.





**Figure A4.** Effects of river discharge. Left column shows final bathymetry of model runs after 2000 yr and the right column shows mud fractions in the top layer of the bed. Run with (a,e) 150 $m^3s-1$, (b,f) 100 $m^3s-1$ (default), (c,g) 50 $m^3s-1$ and (d,h) 0 $m^3s-1$ river discharge.





**Figure A5.** Hydrodynamics and morphology along estuaries with different discharge. From left to right column: model with river discharge of 150, 100, 50 and 0 $m^3s-1$. (a–d) Minimum, mean and maximum bed elevation and high and low water level, (e–h) braiding index, (i–l) estuary width defined as: the initial width, maximum reach over the whole scenario run, the width of wet cells in the model, width defined by a threshold value that is used to mask the cells that are around the dry-wet cell threshold. (m–p) intertidal area and mud cover as percentage of the total area, (q–t) tidal range and (u–x) peak ebb and flood velocities.





**Figure A6.** Hydrodynamics and morphodynamics over time for estuaries with different discharge. From left to right column: model with river discharge of 150, 100, 50 and 0 m³s−1. (a–d) Bathymetry of the cross-section at the mouth plotted over time, (e–h) mud fraction in top layer of cross-section at the mouth, (i–l) estuary width at 1, 4 and 8 km from the mouth, (m–p) funnel-shape parameter, (q–t) estuarine surface area, (u–x) intertidal area and mud in the bed relative to the total area, (y–II) absolute bed level change and (III–VI) net bed level change.





**Figure A7.** Effects of tidal range. Left column shows final bathymetry of model runs after 2000 yr and the right column shows mud fractions in the top layer of the bed. Run with (a,e) 4 m, (b,f) 3 m (default), (c,g) 2 m and (d,h) 1 m tidal range.





**Figure A8.** Hydrodynamics and morphology along estuaries with different tidal ranges. From left to right column: model with 4, 3 (default), 2 and 1 m tidal range. (a–d) Minimum, mean and maximum bed elevation and high and low water level, (e–h) braiding index, (i–l) estuary width defined as: the initial width, maximum reach over the whole scenario run, the width of wet cells in the model, width defined by a threshold value that is used to mask the cells that are around the dry-wet cell threshold. (m–p) intertidal area and mud cover as percentage of the total area, (q–t) tidal range and (u–x) peak ebb and flood velocities.







**Figure A9.** Hydrodynamics and morphodynamics over time for estuaries with different tidal ranges. From left to right column: model with 4, 3 (default), 2 and 1 m tidal range. (a–d) Bathymetry of the cross-section at the mouth plotted over time, (e–h) mud fraction in top layer of cross-section at the mouth, (i–l) estuary width at 1, 4 and 8 km from the mouth, (m–p) funnel-shape parameter, (q–t) estuarine surface area, (u–x) intertidal area and mud in the bed relative to the total area, (y–II) absolute bed level change and (III–VI) net bed level change.





**Figure A10.** Effects of mud source in the presence of waves. Left column shows final bathymetry of model runs after 1250 yr and the right column shows mud fractions in the top layer of the bed. Run with (a,e) only sand, (b,f) marine mud input (default), (c,g) marine and fluvial mud input and (d,h) fluvial mud input.

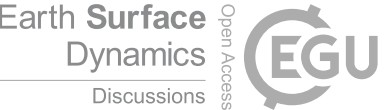



**Figure A11.** Hydrodynamics and morphology along estuaries for different mud sources in the presence of waves. From left to right column: model with only sand, marine mud supply, supply from both boundaries and fluvial supply (default). (a–d) Minimum, mean and maximum bed elevation and high and low water level, (e–h) braiding index, (i–l) estuary width defined as: the initial width, maximum reach over the whole scenario run, the width of wet cells in the model, width defined by a threshold value that is used to mask the cells that are around the dry-wet cell threshold. (m–p) intertidal area and mud cover as percentage of the total area, (q–t) tidal range and (u–x) peak ebb and flood velocities.

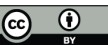



**Figure A12.** Hydrodynamics and morphodynamics over time for estuaries for different mud sources in the presence of waves. From left to right column: model with only sand, marine mud supply, supply from both boundaries and fluvial supply (default). (a–d) Bathymetry of the cross-section at the mouth plotted over time, (e–h) mud fraction in top layer of cross-section at the mouth, (i–l) estuary width at 1, 4 and 8 km from the mouth, (m–p) funnel-shape parameter, (q–t) estuarine surface area, (u–x) intertidal area and mud in the bed relative to the total area, (y–II) absolute bed level change and (III–VI) net bed level change.



*Author contributions.* The authors contributed in the following proportions to concept and design, modelling, analysis and conclusions, and manuscript preparation: LB(40,100,50,70 %), TK(10,0,0,10 %), JL(5,0,5,0 %) and MK(45,0,45,20 %).

*Competing interests.* The authors declare that they have no conflict of interest.

*Acknowledgements.* This work is part of the PhD project of LB in the project 'Turning the Tide' funded by the Dutch Technology Foundation
5 (STW) of the Netherlands Organisation for Scientific Research (NWO), Vici-grant 016.140.316/13710 to MK. We acknowledge Bert Jagers for help with the sand-mud interaction module of Delft3D, Mick van der Wegen for discussions on long-term modelling, MSc thesis student Samor Wongsoredjo for testing boundary conditions for waves and Anne Baar for comments on an earlier version and discussion. Reviewers will be acknowledged.



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
