# Peer review of "Effects of mud supply on large-scale estuary morphology and development over centuries to millennia"

_Earth Surface Dynamics, 2017_

## Referee Comment (RC1) · P. Weill (Referee) · 13 Apr 2017

This study investigates the effects of mud supply on morphodynamics and large-scale morphology of estuaries, and more specifically on the development of mudflats. The adopted approach consists in long-term (2000 years), 2D-horizontal simulations (using Delft3D modelling package) of the evolution of an idealized estuary broadly inspired from the Dyfi estuary in Wales (UK). The authors explore the sensitivity of mudflat development and estuary morphology to boundary conditions (tidal range, waves, fluvial discharge, marine and fluvial SPM concentrations) and sediment-related parameters (settling velocity, active layer thickness, cohesive behaviour of mixtures) through a significant number of simulations (23). Results of the more pertinent simulations are compared by the mean of planform views of the modelled estuaries, and through temporal and spatial analysis of an exhaustive list of morphological and

hydro-sedimentary parameters. The main finding of the study is that estuaries reach a dynamic equilibrium after self-confining, mainly attributed to mudflat formation on the side of the estuaries which reduce channels and bar dynamics. In contrast, pure sand estuaries tend to widen continuously in the absence of cohesion.

Compared to other studies based on morphodynamic modelling of real-world estuaries, this detailed sensitivity analysis on a simple synthetic case provides valuable insights into the role of cohesive sediments on estuarine morphodynamics, or at least into the functioning / behaviour of such morphodynamic numerical models.

**General comments:**

- Spatial pattern of mud flats in estuaries (section 1.1) is introduced through the example of two Dutch estuaries (Western Scheldt and Ems-Dollard), essentially using a compilation of bed samples analyzed in terms of percentage of mud content and hypsometric curves of mud-covered surfaces. The presentation of these examples is quite short, and somehow incomplete as very little information about the hydrodynamics are presented. In my opinion, a more thorough "state-of-the-art" review on the development of mudflats in estuaries, on the evolution of tidal asymmetry during the estuary infilling, and on dynamic equilibrium is missing. Moreover, the authors make little use of these examples in the discussion, as comparison between the simulation results and the Dutch estuaries is mainly qualitative. Quantitative comparisons between simulation results and natural estuaries (in terms of estuarine morphology and hydrodynamcis) are based on datasets of estuaries in UK (Prandle et al., 2005) and around the world (Leuven et al., 2016). Are the Scheldt and Ems-Dollard examples really useful to the paper and discussion ?

- Mainy studies show that estuaries experience a global flood dominance during their earlier infilling history, which may shift to an ebb dominance with changes in hypsometry, in particular with the development of intertidal flats and deeper channels. Simulation results of this study clearly show the ebb-dominance of the equilibrium-state estuary after 2000 years (Fig. 4g,h). Information about the evolution of ebb-flood peak velocity ratios at different times in the estuary development and infilling would bring additionnal elements of discussion to the paper.

- Brown and Davies (2009, 2010) have performed hydro-sedimentary simulations of the Dyfi estuary using the Telemac modelling system, with both natural and idealized bathymetries. They show a clear ebb-dominance in the lower estuary, causing a net seaward sediment transport, which is consistent with the present study. However, they show that tidal flow is flood-dominated in the upper estuary, causing a net transport up-estuary. In the present study, no flood-dominance is observed in the upper estuary, even at low to null fluvial discharge. How do you explain these differences ? I think that the discussion could be improved if the present results are compared to other similar studies. In particular, the validation of the hydrodynamics of the model would strengthen your hypothesis that sediment cohesion is essential to reach a dynamic equilibrium, which can not be solely explained by tidal asymmetry. You could also include and discuss the work of Moore et al. (2009) on the Dee estuary.
[Moore et al. (2009) Morphological evolution of the Dee Estuary, Eastern Irish Sea, UK: A tidal asymmetry approach. Geomorphology 103, 588-596].

**Specific comments:**

- Section 1.1 (p. 2): Presentation of the Western Scheldt and Ems-Dollard estuaries. Some information on hydrodynamics would be useful. What is the tidal range at the estuary mouth ? Tidal excursion ? Tidal current peak velocities ? Global wave climate

?
See for example data presented in Dyer et al. (2000) - An Investigation into Processes Influencing the Morphodynamics of an Intertidal Mudflat, the Dollard Estuary, The Netherlands: I. Hydrodynamics and Suspended Sediment.

- Figure 1. (p. 3) A small location map of the two estuaries would be useful. Please specify the elevation datum used (Amsterdam Ordnance Datum ?)

- Table 1. (p. 6): Please specify in the caption which "sensitive parameters" will be varied (mud settling velocity ?). Please provide the settling velocity of sand, as well as the critical shear stress for erosion of sand.
I do not understand the value given for the critical bed shear stress for sedimentation of mud (1000 N/m$^2$). Generally, the critical shear stress value for deposition is smaller than the one for erosion.
Why are specific densities of mud and sand equal ? 1600 kg/m$^3$ is a typical value for pure sand. However, dry bed density of mud is generally lower than for sand, well below 1000 kg/m$^3$. See for example the data from Wadden Sea sediments from Flemming and Delafontaine (2000), Fig. 2B, where dry bulk density is plotted as a function of mud content. Dry bulk density falls down to 400 kg/m$^3$ for 100% mud content. Other data for different estuaries are presented in Dyer et al. (2000).
[Flemming  Delafontaine (2000) Mass physical properties of muddy intertidal sediments: some applications, misapplications and non-applications. Continental Shelf Research, 20, 1179-1197]
Dyer, Christie, Wright (2000) The classification of intertidal mudflats. Continental Shelf Research, 20, 1039-1060

2. Methods – p.5 l. 10: "(Guo et al., 2016)". See also the work of Moore et al. (2009)

on the Dee estuary (UK).
[Moore et al. (2009) Morphological evolution of the Dee Estuary, Eastern Irish Sea, UK: A tidal asymmetry approach. Geomorphology 103, 588-596]

2.2 Model schematization

- p. 10 l. 2-4. It is not clear how the fluvial discharge is partitioned. Is the sinusoidal partitioning performed between the upstream grid cells of the 300m-wide channel ?

- p. 10 l. 13. How is defined / calculated the flow capacity ? Is it somehow related to the Rouse profile ?

3.1 General development, Figure 3 (p. 13)
- Please change the colour of (a), (b), (c) and (d) labels to white on the bathymetry sub-plots, as the black letters on the purple background is not visible on the printed version.

- In the caption, it is mentioned that bathymetry and mud fraction are shown for simulation times of 50, 150 and 2000 years. However, there are four sub-figures displayed. Please correct the caption. It would also be useful to display the simulation times directly on the upper-right corner of each sub-figure, both for bathymetry and mud fraction.

- This is a very personal preference, but I would prefer $[\times 10^6 m^3]$ instead of $[hm^3]$ in sub-figures (e) and (j).

- It is not clear how are defined the absolute and net bed level changes. Please

provide more details. I am also not convinced by the term "bed level change" as the data correspond to sediment volumes.

- What does the vertical blue dotted lines in sub-figures (e) and (j) represent ? There is not mention of these markers in the manuscript text, nor in the caption, although I suppose these correspond to different phases of adaptation of the model.

- The link to the YouTube video does not work. Please consider submitting the video as supplementary material.

3.2 Hydrodynamics and sediment transport

- p. 14 l. 33: The two formulas defining the tidal prism are "dropped" at the end of the paragraph. I am not sure these are essential. Maybe the tidal prism should be defined simply in a sentence within the paragraph.

- Figure 4, caption: Please specify that negative distances (for instance -10 km) refer to "open sea".

- Figure 4d : The bedload and suspended river curves, supposed to be dashed, appear almost continuous. Please increase space between the dashes.

- Figure 4g,h : Ebb-Flood velocity ratio is positive in sub-figure (g), and negative in sub-figure (h). Ratio values should be consistent between the two plots.

3.3 – "Effects of mud flat formation". I suggest to change this sub-section title by

**ESurfD**
"Effect of mud supply".

p. 16 l. 4: "Locally, mud accretes on bars that are rather stable (e.g. Fig. 3b, on the ebb delta)." Figure 3b shows delta elevation, not mud fraction, and no ebb delta is visible on this image. Please check that Fig. 3b is the figure that should be referred to.

p. 19 Figure 7: Do the data presented in these sub-figures (in particular BI, W, A and Ux) correspond to the final stage of the simulations (i.e. 2000 yr) ? If so, please specify in the figure caption.

p. 19 Figure 7a-d: Please specify somewhere in the caption that the two black dotted lines for initial bed level correspond to the initial bed level of the floodplain (line above zero) and to the intimal bed level of the estuary (line below zero). Figure 7e-h: The braiding index should be defined somewhere in the manuscript, as several definitions exist (Brice 1964, Rust 1978, Howard et al. 1970, Friend Sinha 1993,...), or an appropriate reference should be cited.

p. 20 Figure 8 : Once again, a personal preference, please change $[hm^2]$ to $[\times 10^4 m^2]$ and $[hm^3]$ to $[\times 10^6 m^3]$.

p. 24 Figure 9 : Please suppress the variable units in the caption, as it is already specified in the sub-plot axis labels. "a more aggressive masking technique in which high mud flats are masked": what is a high mudflat ? What are the threshold altitudes for the two methods ?

p. 25 l. 13: "[…] while typically settling velocities of marine mud are significantly

higher." Due to flocculation ? Any reference to support this assumption ?

p. 26 Figure 11: "Lines indicate estimations of estuarine length by Prandle et al. (2005) of 5, 10 and 20 km" add "from left to right".

Throughout the manuscript and in the figures and figure captions, when presenting simulation results, it would be useful to refer directly to run (lab) numbers listed in Table 3.

**Technical corrections:**

- p. 2, l. 7: "large-scale planform that develop over centuries" should read "large-scale planforms that develop over centuries".
- p. 2 l. 11: "Eems-Dollard estuary" should read "Ems-Dollard estuary" (or keep Eems if this is the right spelling, and correct Ems in the rest of the manuscript)
- p. 2 l. 13-14: "so we only look at general patterns and properties" of what ?
- p. 2 l. 14: "measures for mud in the bed" should read "measures of mud in the bed" or "measures of mud content in surficial sediment".
- p. 2 l. 21: "found in areas on the sides and bars shielded from [. . .] tidal flow" should read "found in areas on the sides of bars shielded from [. . .] tidal flow" ?
- p. 3 l.1 (caption): "Eems-Dollard" should read "Ems-Dollard".
- p. 4 l. 16: "less active channels and less channel migration" - redundant ?
- p. 4 l. 20-21: "showed that channel-bar patterns form that are similar to those in nature" should read "showed channel-bar patterns that are similar to those in nature"
- p. 5 l. 6: "to vary the [. . .] boundary conditions for the main question." Please specify "the main question".
- p. 5 l. 21: "(Fischer, 1972, as defined by)" should read "(as defined by Fischer, 1972)"

p. 6 l. 6: "implementation of van Ledden (2001); Jacobs et al. (2011)" should read "implementation of van Ledden (2001) and Jacobs et al. (2011)"
p. 6 l. 9: "it's" should read "its"
p. 6 l. 16: ",but" should read ", but" (add space)
p. 7 l. 19: "0.25 mm/s" change to "0.25 mm s$^{-1}$"
p. 7 l. 21 "0.1 - - 0.4 mm s$^{-1}$" should read "0.1 - 0.4 mm s$^{-1}$"
p. 7 l. 24-25 : "similar to Le Hir et al. (2011); Sanford (2008)." should read "similar to Le Hir et al. (2011) and Sanford (2008)."
p. 8 l. 21: "bedslope" should read "bed slope"
p. 9 l. 3: "between de" should read "between the"
p. 9 l. 15: "The sea has an depth" should read "The sea has a depth"
p. 9 l. 15: "van der Wegen" should read "Van der Wegen"
p. 10 l. 2: "$100m^{-3}s^{-1}$" should read "$100m^3s^{-1}$"
p. 10 l. 17: "20 d" change to "20 days"
p. 11 Table 3, header of column 5 and 8: "$s - 1$" should read "$s^{-1}$" (superscript)
p. 11 l. 14: "bedslope" should read "bed slope"
p. 12 l. 8: "of20 mgL$^{-1}$" should read "of 20 mgL$^{-1}$" (add space)
p. 13 l. 18: "the an initial bar pattern" should read "an initial bar pattern"
p. 14 l. 3-4: "At the mouth the water level rapidly progresses from low to high water and slowly progresses from high to low water." Replace by "At the mouth the water level rapidly increases from low to high water and slowly decreases from high to low water."
p. 14 l. 10: "around7.5 km" should read 'around 7.5 km" (add space)
p. 14 l. 20: SPM acronym has not been defined before (Suspended Particulate Matter)
p. 14 l. 20: "45 mg/l" should read "45 mg L$^{-1}$"
p. 16 l. 6: "mud- dominated" should read "mud-dominated" (delete extra space)
p. 16 l. 6-7: Please rephrase sentence. Proposition : "As a consequence, the critical shear stress for sand erosion equals the entrainment threshold of mud (Eq. 6). The mud-dominated mixed sediment thus becomes more difficult to erode and more rapid

aggradation is likely to occur".
p. 16 l. 29-31: Too many "furthermore" and "therefore"
p. 18 caption: "at below this elevation" should read "below this elevation"
p. 22 l. 22-23: "higher flood velocities near the mouth, especially flood velocities" should read "higher flow velocities near the mouth, especially flood velocities" ?
p. 22 l. 23-24: Please rephrase the end of the sentence.
p. 22 l. 26: "The sensitivity [. . .] was assessed by" should read "The sensitivity [. . .] assessed by"
p. 22 l. 33-34: Please rephrase the sentence (difficult to read and understand)
p. 24 caption l. 2: "mouth with" should read "mouth width"
p. 25 l. 6: "McLaren dataset" should read "McLaren (year) dataset"
p. 27 l. 4: "absence or river influence" should read "absence of river influence"
p. 27 l. 12: "mud flat with at the mouth" should read "mud flat width at the mouth"
p. 28 l. 29: "a range of tidal amplitudes and river discharge" should read "a range of tidal amplitudes and river discharges"
p. 33 caption: "$m^3s-1$" should read "$m^3s^{-1}$" (superscript)
p. 34 caption: "$m^3s-1$" should read "$m^3s^{-1}$" (superscript)
p. 35 caption: "$m^3s-1$" should read "$m^3s^{-1}$" (superscript)
p. 43 l. 3-4: Please check the reference. "Team, C." is "CCCR Team". This reference is a conference poster. Please provide the conference details: BSRG 2008, December 14th-17th, Liverpool.
p. 43 l. 31: "Journal of fluid mechanics" should read "Journal of Fluid Mechanics"
p. 46 l. 28: "betweenvegetation" should read "between vegetation" (missing space)

---

## Referee Comment (RC2) · Anonymous Referee #2 · 27 May 2017

This paper by Braat et al. presents 2DH numerical experiments of idealized estuaries in order to investigate the effect of mud supply on the long term development of estuaries. Several similar studies were already performed for sandy estuaries and deltas but here the originality of this study is to consider both sand and mud. I think that this study matches well the topics usually addressed in E-surf and that it would be a worth contribution in the literature. The paper is well organized and written, the figures are clear, the predicted final morphologies are impressive, the literature cited is relevant and I think that the paper would only need moderate revisions before it can be accepted. Please note that the following relatively long list of comment is related to the length of the paper. This review is split into moderately important problems that concern the whole paper and relatively minor, along-the-text problems.

Moderately important problems:

[Figure]

-Use of morphological factors. While I recognize that there is presently no alternative to morfac approaches to perform morphological simulations at millennia time scales, such approaches were never applied for sand and mud mixtures and it is not straightforward that the underlying assumptions are valid. I suggest that the authors perform a few runs over the longest period that they can (at least 5 years) and compare the final morphology obtained without morfac with that obtained with morfac of e.g. 10, 100 and 400. A figure presenting these results could be included as a new appendix and discussed in the beginning of the discussion in a section untitled "limitations of the modeling approach" or something like that. I think this would strengthen the present paper and be useful for future modeling studies.

-Development of bended channels. The model is apparently able to reproduce the development of bended channels, which is by the way noted by the authors P12, L.17 and P22, L16. However, a 2DH modelling system cannot represent the vertical circulation that takes place in bended channel and the subsequent sediment transport. Is there a special treatment in Delft3D to account for this process in 2DH as it is often the case in river morphodynamic studies?

-Section 3.6 is clearly a discussion subsection and should be moved to the discussion. As much as possible, discussion should be limited in the "Results" section and moved to the discussion section.

Along the text, minor comments:

-P1, L4: "estuaries".

-P1, L17: some estuaries are also dominantly built of mud. Find a reference or clarify.

-P2, L2: which conditions?

-P3, L2: past modelling of what? Le Hir et al., (2011) and I'm sure others already performed sediment transport simulations with sand and mud mixtures, please be more careful with "always". For instance, the authors missed a couple of paper by Geleynse

(e.g. Geleynse et al., 2011) where Delft3D is applied to idealized deltas and where both sand and silts are considered. These studies might also be considered for the discussion in section 3.2.

-P3, L13: again, please consider the series of studies by Geleynse et al. (e.g. 2011).

-P5, L5: This is somehow confusing to introduce the EMS Estuary in section 1.1 and now move to the Dovey Estuary. Please better justify why you considered these two examples.

-P5, L15: please rather use "modelling system" than "simulation program".

-P5, L24: if Delft3D is used in 2DH, then the Saint-Venant equations are solved, which correspond to the depth integrated Navier-Stokes equations. Also, not that, as written, eqs. (1) to (3) do not represent the effect of short waves.

-P7, eq. (4): how is solved the Exner equation to compute bottom change from the divergence of bedload transport? How are treated transition zones between where $Pm<Pm,cr$ and $Pm>Pm,cr$?

-P7, eq (5): Tau,cw is not explained in the text.

-P7, L23: the gradient of a vector is only the same as its divergence in 1D. In 2DH, bed level changes are caused by the divergence of sand fluxes.

-P7, L27-29: this sentence is not straightforward, please explain a bit better.

-P8, Table2: "transverse slope parameter", do you mean a slope limiter?

-P9, L8: "rectilinear non-uniform" is a pleonasm as a uniform rectilinear grid is a regular grid. I would rather say "a rectilinear grid, which resolution ranges from WW and XX".

-P9, L13: 3 km-wide.

-P9, L15: a depth. Is that the maximum depth?

-P9, L30: "a M2 tide". Then "3° phase difference" with respect to what? Between the

seaward point and the shore so that tide is shore-normal? What about the Western Boundary? Why not prescribing tides?

-P10, L5-9: with such a coarse resolution and small waves, wave-induced processes cannot be represented properly. As a rough guideline, the grid should have a least 5 elements across the surfzone to represent properly wave-induced currents and setup. Here I assume that only wave stirring of sediment is represented in the model, and possibly a slight increase in bed shear stress. Please verify and clarify.

-P10, L17: why not using Delft3D in parallel? Parallel computing is common practice nowadays and would allow to use smaller morfac for instance or have your final results in <1 day instead of 20.

-P14, L4: there is indeed a phase difference of Pi/2 between water levels and velocities. Do you mean that this phase lag does not vary too much along the estuary? This is not that clear on figure 4.

-P14, L28: there are no dotted lines on figure 4.

-P14, L32: what do you mean by "not very large"? Do you mean that this is different from the 7.5% provided above?

-P14, L33: please explicit "w".

-P21, section 3.4: this is only a thought but is that possible that an estuary that imports mud from the sea has no mud import from the river?

-P22, L12-13: is that realistic that the estuary closes in the absence of waves? I think that in reality, the only estuary that are closed are wave-dominated.

-P22, L21-24: this is not clear at all why waves would rise high water level and increase tidal range. According to previously published studies (e.g. Wargula et al., 2014; Dodet et al., 2013), wave breaking on the ebb shoal rises the water level in the estuary/lagoon by about 10% of the significant wave height at breaking. Since surf zone can hardly be

represented with the resolution employed in this study, I don't see how waves can have any effect other than stirring sediments and, marginally, increase bed shear stress. Please clarify this section.

-P24, figure 9: how can an estuary without mud supply have large surface areas covered by mud?

-P26: why Gg.yr-1 and not 10ˆ6 kg/yr?

-P26, L10: use of "which" not correct I think.

-P27, table 4: how river discharge can induce currents in the range 0.001-0.01 m/s only? Upstream, it could be much more than that? Why weren't you able to compute it from the model, for instance based on a run with river discharge only?

- P28, L15-19: in reality, tides big enough to develop estuaries imply that the associated oceanic basin is large enough to have significant short-waves as well. Short waves tend to limit ebb-dominance and subsequent estuary enlargement. If required, you'll find a review in the introduction of Wargula et al. (2014).

Cited literature:

Dodet, G., Bertin, X., Bruneau, N., Fortunato, A.B., Nahon, A., Roland, A., 2013. Wave-current interactions in a wave-dominated tidal inlet. Journal of Geophysical Research: Oceans, 118 (3), pp. 1587-1605.

Geleynse, N., Storms, J.E.A., Walstra, D.J.R., Jagers, H.R.A., Wang, Z.B., Stive, M.J.F., 2011. Controls on river delta formation; insights from numerical modeling. Earth and Planetary Science Letters, 302 (1-2), pp. 217-226.

Wargula, A., Raubenheimer, B., Elgar, S., 2014. Wave-driven along-channel subtidal flows in a well-mixed ocean inlet. Journal of Geophysical Research: Oceans, 119 (5), pp. 2987-3001.

**ESurfD**

Interactive
comment

---

## Author Comment (AC1) · 23 Jun 2017

We thank Pierre Weill for the detailed review and careful reading of the manuscript which helped us to greatly improve the manuscript.

The first general comment is that the connection between the results and the presented examples of two Dutch estuaries needs more explanation. The dataset of UK estuaries is used quantitatively to compare hydrodynamics and bar shape and size, while the Dutch estuaries are mainly used qualitatively for comparison of mud deposition patterns. This is mainly because data on surface mud in estuaries is scarce and unavailable for the UK estuaries. The Dutch data shows spatial distribution of the mud over an estuary, which is relevant because it validates the spatial patterns obtained in

our model results, so there is more similarity than only planform and hydrodynamics. We will discuss this in more detail in the resubmission and also compare the hypsometric curves for the field and modelled cases. Yet, we chose to model the Dovey instead of the Western Scheldt or Ems-Dollard because we wanted to model a largely natural estuary. The Dutch estuaries are largely influenced by dredging and dumping, which for the modelling would require significant unnatural volume displacement. In the next version of the manuscript we will add relevant hydrodynamic information of these estuaries. Furthermore we will explain more clearly why we model the Dovey in the beginning of the method section. We will also indicate the positions of the Dutch estuaries in the figure with the UK and other data that serves as context for the modelling.

The second and third general comments request more discussion on ebb- and flood dominance, in particular through time which would be novel relative to earlier work in the Dovey estuary. We will show ebb-flood peak velocity ratios over time in the resubmission and include a discussion, comparing our results to the insights of Moore et al. (2009) and Brown and Davies (2009,2010 ). We foresee an important difference in that ebb/flood asymmetry works different for sand and for mud, meaning that an estuary can export sand and import mud at the same time.

In response to the specific comments we added more information on the Western Scheldt and Ems-Dollard estuary. We also changed Fig. 1 by adding a small location map and the elevation datum as suggested. The caption of Table 1 was clarified and settling velocity for sand was provided.

Other questions were asked by the reviewer regarding the high critical shear stress for deposition and the dry bed density of mud in Table 1. A discussion on the subject of the deposition threshold can be found in Sanford (2008) and several papers of Winterwerp and a reference to this will be added in the new manuscript. Many papers suggest that an erosion threshold for deposition is absent and should be determined by: D=ws*c. This is approached in the model by setting a very high value for the critical shear stress for sedimentation, which is common practice in Delft3D with mud. We chose for equal

dry bed densities for sand and mud because the density of mud is highly variable. The density is indeed much lower when mud is recently deposited on for example a mudflat (as in Flemming & Delafontaine, 2000, and Dyer et al., 2000, where samples are taken from the active layer), but consolidation is very strong and rapid, as soon as the mud becomes buried the density becomes higher than for sand. In principle the density can be anywhere on the line in figure 6 of Flemming & Delafontaine (2000). Since Delft3D did not account for consolidation during the modelling phase of this research we chose the dry density to be equal to sand as a conservative estimate. Consolidation was already mentioned as a discussion point later in the paper and we consider this an important uncertainty in the model. Meanwhile, Delft3D has developed a consolidation module, which we will use in future mud-modelling research.

In the method section we now also refer to Moore et al. (2009) in relation to testing certain tidal components. Furthermore, we clarified in the methods section how fluvial discharge is partitioned and flow capacity is defined.

In the results the labels, caption and red lines in Fig. 3 have been adapted and clarified as suggested. Definitions of absolute and net bed level change ware added in the manuscript. However, we abstained from using $\times 10^6$ m3 instead of hm3 further in the manuscript because it saves space in the figure. Additionally, we did not experience any problems with the YouTube link and therefore remains the same.

p.14 l.33: Tidal prism is now described in words in the paragraph instead of the equation.

Regarding Figure 4: The coordinates are already explained in the caption of the method figure that shows the initial bathymetry. Dashes are changed as is the vertical scale in subfigure (g).

Section title 3.3 was changed from "Effects of mud flat formation" to "Effects of mud supply".

p. 16 l. 4: The reference to the figure has been adapted.

p. 19 Figure 7: Caption has been changed as for the similar figures of other model runs in the appendix.

p. 24 Figure 9: Variable units are suppressed and 'high mudflat' is explained in caption.

p. 25 l. 13: Reference of Leussen (2011) and Mietta (2009) were added to support assumption of higher settling rates of marine mud related to flocculation processes.

p. 26 Figure 11: added "from left to right" in caption.

We added model numbers and references to appendix figures throughout the manuscript text and captions. The model numbers are related to Table 3.

Finally, all technical corrections have been implemented in the manuscript as well.

References:

Brown, J. M., & Davies, A. G. (2009). Methods for medium-term prediction of the net sediment transport by waves and currents in complex coastal regions. Continental Shelf Research, 29(11), 1502-1514.

Brown, J. M., & Davies, A. G. (2010). Flood/ebb tidal asymmetry in a shallow sandy estuary and the impact on net sand transport. Geomorphology, 114(3), 431-439.

Dyer, K. R., Christie, M. C., & Wright, E. W. (2000). The classification of intertidal mudflats. Continental Shelf Research, 20(10), 1039-1060.

Flemming, B. W., & Delafontaine, M. T. (2000). Mass physical properties of muddy intertidal sediments: some applications, misapplications and non-applications. Continental Shelf Research, 20(10), 1179-1197.

van Leussen, W. (2011). Macroflocs, fine-grained sediment transports, and their longitudinal variations in the Ems Estuary. Ocean Dynamics, 61(2-3), 387-401.

Mietta, F., Chassagne, C., Manning, A. J., & Winterwerp, J. C. (2009). Influence of

shear rate, organic matter content, pH and salinity on mud flocculation. Ocean Dynamics, 59(5), 751-763

Moore, R. D., Wolf, J., Souza, A. J., & Flint, S. S. (2009). Morphological evolution of the Dee Estuary, Eastern Irish Sea, UK: a tidal asymmetry approach. Geomorphology, 103(4), 588-596.

Sanford, L. P. (2008). Modeling a dynamically varying mixed sediment bed with erosion, deposition, bioturbation, consolidation, and armoring. Computers & Geosciences, 34(10), 1263-1283.

---

## Author Comment (AC2) · 23 Jun 2017

We would like to thank anonymous referee 2 for the constructive review. The comments related to the description of the model help to make the paper much clearer.

The first comment drew our attention to the point that long term morphological simulations with a morphological factor and mud were never tested before in the literature. However, during the earlier modelling stages of this research we did some tests with other morphological acceleration values (10 to 1000) with only sand and different types of morphological acceleration factors, for example related to the sedimentation and erosion of one tidal cycle. We initiated this because we found it problematic that by using a morphological factor in tidal systems the tidal half-period over which sediment is trans-

ported in one direction is basically multiplied as well. Nevertheless, with these tests we found that the most widely used morphological factor we now use in the paper is actually the least different from runs with lower factors in these types of models (Roelvink, 2006 and pers. comm. 2016). In the next manuscript two additional model runs with lower morphological acceleration factors will be included as an appendix.

For the development of bended channels in the 2DH model, the second comment, we indeed used a parameterisation for the deflection of the bed shear stress vector depending on the spiral flow that is calculated from local curvature. We added this to the methods section under numerical model description.

The third comment suggests that section 3.6 should be moved to the discussion. We agree that this section includes some interpretation, but we think this subsection includes new results that were not described in earlier paragraphs. Furthermore, the results are not connected to any literature or larger context. Therefore, we would like to leave this section at the end of the results just before the discussion.

In the next part, italic sentences are comments of the reviewer that we want to address individually.

*P3, L2: Le Hir et al., (2011) and I'm sure others already performed sediment transport simulations with sand and mud mixtures, please be more careful with "always". For instance, the authors missed a couple of paper by Geleynse (e.g. Geleynse et al., 2011) where Delft3D is applied to idealized deltas and where both sand and silts are considered. These studies might also be considered for the discussion in section 3.2.* - We specified the situation where we used always. The paper of Geleynse et al. (2011) does not contain mud, and is therefore less relevant than the other delta papers that were mentioned. In this paper we focus on estuaries, but because limited studies were done on estuaries with mud we added a few delta papers that work with mud.

*P5, L24: if Delft3D is used in 2DH, then the Saint-Venant equations are solved, which correspond to the depth integrated Navier-Stokes equations. Also, not that, as written,*

*eqs. (1) to (3) do not represent the effect of short waves.* - We believe Saint-Venant equations are 1D, not 2D. We will refer equation (1) to (3) as the shallow water equation in the next manuscript. The effect of short waves is solved separately in SWAN and this effect is therefore not included in these equations.

*P7, eq. (4): how is solved the Exner equation to compute bottom change from the divergence of bedload transport? How are treated transition zones between where $Pm<Pm,cr$ and $Pm>Pm,cr$?* - We will clarify the bottom change. The transition zone of the critical mud content is quite abrupt as explained in last paragraph of section 2.1.

*P8, Table2: "transverse slope parameter", do you mean a slope limiter?* - No, these are the parameters for equation 7 in which sediment transport on transverse slopes is deflected as a function of slope and sediment mobility.

*P10, L5-9: with such a coarse resolution and small waves, wave-induced processes cannot be represented properly. As a rough guideline, the grid should have a least 5 elements across the surfzone to represent properly wave-induced currents and setup. Here I assume that only wave stirring of sediment is represented in the model, and possibly a slight increase in bed shear stress. Please verify and clarify.* - Correct, we only used the waves for wave stirring to bring the mud in suspension; we clarified this more clearly in the method section. Initially we wanted to simplify the model by ignoring waves in all model runs, but the mud deposition at the borders (with a marine source of mud) generated instabilities as visible in Fig. A1. Therefore, it was necessary to add wave stirring to prevent deposition in these locations. Properly modelled wave-induced transport is very difficult to use in combination with the Engelund-Hansen transport equation that we use now. Wave processes are more commonly used in combination with the Van Rijn transport equations, but these equations produce less realistic patterns in long-term morphological simulations. Basically, we did not want to sacrifice bar-channel pattern for realistic wave transport.

*P14, L4: there is indeed a phase difference of Pi/2 between water levels and velocities.*

**ESurfD**
*Do you mean that this phase lag does not vary too much along the estuary? This is not that clear on figure 4.* - I think there is no phase lag, because when water levels are maximal or minimal, the velocity is exactly zero as you would expect without a phase lag.

*P21, section 3.4: this is only a thought but is that possible that an estuary that imports mud from the sea has no mud import from the river?* - The reason for this scenario is to develop understanding of sediment provenance. However, it is also realistic as one could see this as a case with an upstream dam which is a common cause of starvation of deltas and saltmarshes. The use of this scenario will be clarified in section 3.4.

*P22, L12-13: is that realistic that the estuary closes in the absence of waves? I think that in reality, the only estuaries that are closed are wave-dominated.* - We agree, that the formation of a spit by waves is the most likely reason that an estuary will close, but if the velocities generated by the tidal flow are very low and enough sediment is available closure without waves should be possible as well. We interpret our modelling to show there is a continuum from river-dominated estuaries to deltas, where the transition to a delta means that the estuary is mostly closed except for a channel with approximately the same width as the upstream river.

*P22, L21-24: this is not clear at all why waves would rise high water level and increase tidal range. According to previously published studies (e.g. Wargula et al., 2014; Dodet et al., 2013), wave breaking on the ebb shoal rises the water level in the estuary/lagoon by about 10% of the significant wave height at breaking. Since surf zone can hardly be represented with the resolution employed in this study, I don't see how waves can have any effect other than stirring sediments and, marginally, increase bed shear stress. Please clarify this section.* - We will look into the cause of these higher water levels. We cannot give a good explanation yet. For now it seems that the waves are not directly the cause but it is an effect of morphology caused by the waves because the difference does not occur at the beginning of the model.

*P24, figure 9: how can an estuary without mud supply have large surface areas covered by mud?* - This is not the case. There is probably some confusion about the blue and brown lines. The brown lines indicate mud-related parameters, blue lines are related to the shape and size of the estuary as explained in the caption. In (d) and (g) the brown line starts at zero. We checked whether the text in the manuscript was clear.

*P28, L15-19: in reality, tides big enough to develop estuaries imply that the associated oceanic basin is large enough to have significant short-waves as well. Short waves tend to limit ebb-dominance and subsequent estuary enlargement. If required, you'll find a review in the introduction of Wargula et al. (2014 ).* - We will add the effect of waves on ebb-dominance as a point of attention when interpreting the results.

Finally, all other minor comments were implemented in the new version of the manuscript.

References:

Geleynse, N., Storms, J. E., Walstra, D. J. R., Jagers, H. A., Wang, Z. B., and Stive, M. J. (2011). Controls on river delta formation; insights from numerical modelling. Earth and Planetary Science Letters, 302(1), 217-226.

Roelvink, J. A. (2006). Coastal morphodynamic evolution techniques. Coastal Engineering, 53(2), 277-287.

---

## Author Response (AR1)

What follows is the track changes document of the manuscript. The manuscript was adapted as described in the author's comments (AC1 and AC2) of the interactive discussion. Some figures have been adapted and are included at the end of the document.

%% Copernicus Publications Manuscript Preparation Template for LaTeX Submissions
%% --------------------------------
%% This template should be used for copernicus.cls
%% The class file and some style files are bundled in the Copernicus Latex Package which can be downloaded from the different journal webpages.
%% For further assistance please contact the Copernicus Publications at: publications@copernicus.org
%% http://publications.copernicus.org

\documentclass[esurf, manuscript]{copernicus}

\begin{document}

\title{Effects of mud supply on large-scale estuary morphology and development over centuries to millennia}

% \Author[affil]{given_name}{surname}
\Author[1]{Lisanne}{Braat}
\Author[2]{Thijs}{van Kessel}
\Author[1]{Jasper R.F.W.}{Leuven}
\Author[1]{Maarten G.}{Kleinhans}

\affil[1]{Utrecht University, Heidelberglaan 2, 3584 CS Utrecht, the Netherlands}
\affil[2]{Deltares, Boussinesqweg 1, 2629 HV Delft, the Netherlands}

\runningtitle{Effects of mud supply on large-scale estuarine morphology}
\runningauthor{Lisanne Braat}
\correspondence{Lisanne Braat (L.Braat@uu.nl)}

\received{}
\pubdiscuss{} %% only important for two-stage journals
\revised{}
\accepted{}
\published{}
%% These dates will be inserted by Copernicus Publications during the typesetting process.

\firstpage{1}

\maketitle

\begin{abstract}
Alluvial river estuaries consist largely of sand but are typically flanked by mud flats and salt marshes. The analogy with meandering rivers, that are kept narrower than braided rivers by cohesive floodplain formation, raises the question how large-scale estuarine morphology and late Holocene development of estuaries are affected by cohesive sediment. In this study we combine sand and mud transport processes and study their interaction effects on morphologically modelled  estuaries on centennial to millennial time-scales. The numerical modelling package Delft3D was applied in 2DH starting from an idealised convergent estuary. The mixed sediment was modelled with an active layer and storage module with fluxes predicted by the Partheniades-Krone relations for mud, and Engelund-Hansen for sand. The model was subjected to a range of idealised boundary conditions of tidal range, river discharge, waves and mud input. The model results show that mud is

predominantly stored in mudflats on the side of the estuary. Marine mud supply only influences the mouth of the estuary whereas fluvial mud is distributed along the whole estuary. Coastal waves stir up mud and remove the tendency to form muddy coastlines and the formation of mudflats in the downstream part of the estuary. Widening continues in estuaries with only sand while mud supply leads to a narrower constant width and reduced channel and bar dynamics. This self-confinement eventually leads to a dynamic equilibrium where lateral channel migration and mud flat expansion are on average balanced. However, for higher mud concentrations, higher discharge and low tidal amplitude the estuary narrows and fills to become a tidal delta.
\end{abstract}

%short summary: Mud plays a critical role in ecological restoration measures and harbour maintenance in estuaries, but is rarely taken into account in numerical morphological models. Starting from the idea that rivers self-constrain their dimensions through cohesive vegetated floodplain formation, which then determines bar pattern, we here study whether mud in estuaries has a similar effect. This is a novel question in the literature.
%The additional novelty of this article is that we add mud supply to long-term morphological models of estuaries over centuries to millennia. Such long timescales are rarely studied in general and to the authors knowledge have never been studied including mud.

%short summary (non jargon): Mud affects the shape and dynamics of estuaries, but is rarely studied in models even though we regularly deal with mud in the short term management of estuaries. We discovered that over centuries to millennia mud supply confines the estuary by mudflats on the sides, while estuaries with only sand continue to grow. Also, the shifting of channels and bars is decreased by the addition of mud. This implies that changes in mud supply in estuaries can lead to changes in shape and dynamics over time.

\introduction
Sandy river estuaries with continuously migrating channels and bars have great and often conflicting economic and ecologic values. These estuaries are typically dominantly built of sand, but mud and salt marshes also form significant parts of these systems. Mud plays a critical role in ecological restoration measures and harbour maintenance, but is rarely taken into account in numerical morphological models. Due to human interference mud concentrations have increased far above the desired values in many estuaries \citep{winterwerp2011,maren2016}. Mud problems arise from pollutants attached to clay particles, mud deposits covering benthic species, rapidly silting harbours and channels and changing hydro- and morphodynamic conditions by higher resistance against erosion. This raises questions about effects of mud on large-scale estuary morphology in natural alluvial systems as a control for cases with human interference.

In rivers, formation of cohesive floodplains with mud and vegetation causes river channels to be narrower and deeper than in systems with only sand given otherwise equal conditions \citep{tal2007,kleinhans2010,dijk2013,schuurman2016}. This results from a dynamic balance between floodplain erosion by migration of channels and new floodplain formation by mud sedimentation and/or vegetation development. The effective cohesiveness may change an unconfined braided system into a dynamic self-confined meandering system or even a straight, laterally immobile channel without bars \citep{makaske2002,kleinhans2011}. Here we study whether mud has similar effects on large-scale planform_s that develop over centuries to millennia in estuaries. We especially need more knowledge about where mud deposits occur and how they influence the evolution of the estuary over long timescales. We first quantify mud flat properties in two Dutch estuaries and then review approaches to mud modelling.

\subsection{Spatial pattern of mud flats in estuaries}

In this study we use data from two Dutch estuaries, the Western Scheldt estuary and the Ems-Dollard estuary The Western Scheldt is a meso- to macrotidal estuary with a semi-diurnal tide and is located in the southwest of the Netherlands (Fig.~\ref{datafigure}f). The estuary has a tidal prism of $2 \times 10^9~\unit{m^3}$ and maximum channel velocities are in the order of $1-1.5~\unit{m s^{-1}}$ \citep{wang2002}. The fresh water discharge is on average $120~\unit{ m^3s^{-1}}$ from the Scheldt River. The Ems-Dollard is a mesotidal estuary with a semi-diurnal tide and is located at the most northern part of the border between Germany and the Netherlands (Fig.~\ref{datafigure}f).  The estuary has a tidal prism of $1 \times 10^9\unit{m^3}$ and maximum channel velocities are in the order of $1~\unit{m s^{-1}}$ \citep{dyer2000}. Fresh water input comes from the Ems River with an average discharge of $80~\unit{m^3s^{-1}}$. We use these estuaries because they are relatively well documented, although bed composition data is rather scarce compared to bed elevation scans. The disadvantage of data in a well-studied estuary is that anthropogenic influences are usually considerable, so we only look at the general patterns and properties of the mud. Here we combine independent measures  of mud content in surficial sediment: 1)~a bed sampling dataset of the Western~Scheldt \citep[Fig.~\ref{datafigure}a;][]{mclaren1993,mclaren1994}, 2)~probability of clay in the GeoTOP~map~(v1.3) of interpolated borehole data in the top $50~\unit{cm}$ of the bed \citep{tno2016} (Fig.~\ref{datafigure}b and e), where clay is defined as more than $35~\unit{\%}$ lutum ($<2~\unit{\mu m}$) and less than $65~\unit{\%}$ silt ($<63~\unit{\mu m}$) \citep{vernes2005}, 3)~yearly Western~Scheldt ecotope maps of \citet{rws2012}, in particular the mud-rich areas above low water level (Fig.~\ref{datafigure}c), that are based on aerial photographs, and 4)~the sediment atlas of the Waddensee \citep{atlas} drawn from bed sampling in 1989 \citep{heuvel1991} which includes the Ems-Dollard (Fig.~\ref{datafigure}d).

Data from the two estuaries indicate that mud is deposited on the sides of the estuary (Fig.~\ref{datafigure}a--d) shielded from the strongest tidal flow and larger fractions of mud are  also found on bars in general  with the estuarine facies description of \citet{dalrymple2007}. The hypsometric curves (Fig.~\ref{datafigure}g and h) indicate that most of the mud is deposited on the intertidal area, yet, significant fractions are also found in channels. Additionally larger mud fractions are observed in the single-channel upper estuaries (Fig.~\ref{datafigure}a, d and f).  To summarise, $10-20~\unit{\%}$ of the lower estuary cross-section is typically covered by mud, increasing to higher fractions up to about half the cross-section in the single-channel upper estuary.

\begin{figure*}[t]
\includegraphics[width=17cm]{mix}
\caption{Mud in the bed of the Western~Scheldt and the Ems-Dollard. (a)~Percentage of mud in the top $10~\unit{cm}$ of the bed \citep{mclaren1993,mclaren1994}, (b)~GeoTOP map (v1.3) of probability of clay in the top $50~\unit{cm}$ of the bed \citep{tno2016}, (c)~indicative morphodynamics map of the Western Scheldt \citep{rws2012}. (d)~Fraction of mud in the top $10~\unit{cm}$ of the bed \citep{heuvel1991,atlas}, (
[revised manuscript text omitted]
 ($\unit{m s^{-1}}$), $g$~is gravitational acceleration ($\unit{m s^{-2}}$), $C$~is the Chezy friction parameter ($\unit{m^{0.5} s^{-1}}$) and $v_{w}$~is the eddy viscosity ($\unit{m^{2} s^{-1}}$).

A module was recently developed for mixed sediments which incorporates the effect of bed composition on erosional behaviour and hence morphology \citep{kessel2011,deltares2012}. This module is a partial implementation of \citet{ledden2001b,jacobs2011} and \citet{jacobs2011} and tracks spatial and temporal bed composition for multiple grain sizes of sand and mud with erosional characteristics depending on bed composition. In this paper we only used one sand fraction and one mud fraction (Table~\ref{sediment}) and  applied a uniform roughness.

\begin{table*}[t]
\caption{Sediment characteristics applied in the default model. Variation in settling velocity will later be discussed as one of the Ssensitivitye parameters will be varied in scenarios discussed later.}
\begin{tabular}{r l c l}
Sediment property & symbol & value & unit \\
        \tophline
        Sand \\
        \middlehline
        Settling velocity & $w_{s}$ & $4.4 \times 10^{-2}$ & $m/s$ \\
        Median grain size & $D_{50}$ & $3 \times 10^{-4}$ & $m$ \\
        Specific density & $\rho_{s}$ & $2650$ & $kg/m^3$ \\
        Dry bed density & $\rho_{dry}$ & $1600$ & $kg/m^3$ \\
        \middlehline
        Mud \\
        \middlehline
        Settling velocity & $w_{s}$ & $2.5 \times 10^{-4}$ & $m/s$ \\
        Critical bed shear stress for sedimentation & $\tau_{crit,sed}$ & $1000$ & $N/m^2$ \\
        Critical bed shear stress for erosion & $\tau_{crit,ero}$ & $0.2$ & $N/m^2$ \\
        Erosion parameter & $M$ & $1 \times 10^{-4}$ & $kg/m^2/s$ \\
        Specific density & $\rho_{s}$ & $2650$ & $kg/m^3$ \\
        Dry bed density & $\rho_{dry}$ & $1600$ & $kg/m^3$ \\
        \bottomhline
\end{tabular}
\label{sediment}
\end{table*}

Cohesive sediment, i.e. mud, is defined as the mixture of the clay ($<2$~\unit{\mu m}$) and silt ($2-63$~\unit{\mu m}$) fractions, where it's cohesive behaviour is mainly caused by physico-chemical forces between the clay particles. This cohesive behaviour causes complex processes that influence erosion and deposition of sediments. In the model we distinguish two erosion modes. Above a critical mud content ($p_{m,cr}$) of the bed, sand particles are not in direct contact but are covered by cohesive particles, which limits erosion for both sand and mud \citep{torfs1995,torfs1996}. Below this critical mud content, friction and gravity oppose sediment transport for sand. The critical mud

content was chosen to be at a mass fraction of $0.4$, which depends on site specific silt-clay ratios because only the clay fraction is cohesive \citep{mcanally2001, ledden2004}. This value is higher than found in flume experiments ($0.1-0.2$, \citeauthor{torfs1995} \citeyear{torfs1995}; $0.05-0.15$, \citeauthor{torfs1996} \citeyear{torfs1996}; $0.02-0.15$, \citeauthor{mitchener1996} \citeyear{mitchener1996}), but was based on silt-clay ratios of Dutch tidal systems \citep[$0.25-0.5$,][]{ledden2004b}.

When the bed is defined as non-cohesive ($p_{m} < p_{m,cr}$), a traditional sand transport equation was used. Here we chose the Engelund and Hansen transport equation (\citeyear{eh1967}; Eq.~\ref{eq:eh}):
\begin{equation}
\ q_{s} = \frac{0.05 U^{5}} {\sqrt{g} C^{3} \Delta^{2} D_{50}} \label{eq:eh}
\end{equation}
where $q_{s}$~is sediment transport ($\unit{m^3 m^{-1} s^{-1}}$), $U$~is the magnitude of the flow velocity ($\unit{m s^{-1}}$), $\Delta$~is the relative density ($\rho_{s}-\rho_{w})/\rho_{w}$ and $D_{50}$~is the median grain size ($\unit{m}$). This equation does not distinguish between suspended and bedload transport, but considers total transport.

The erosion rate of mud was calculated by the Partheniades-Krone formulation \citep[][Eq.~\ref{eq:PK1}]{partheniades1965}:
\begin{equation}
\ E_{m} = MS(\tau_{cw},\tau_{cr,e}) \label{eq:PK1}
\end{equation}
where $E_m$~is the erosion flux of mud ($\unit{kg m^{-2} s^{-1}}$), $M$~is the erosion parameter ($\unit{kg m^{-2} s^{-1}}$), $S$~is the erosion or depositional step function, $\tau_{cr,e}$~is critical shear stress for erosion ($\unit{N m^{-2}}$), and $\tau_{~cr,dcw}$~is the maximum bed shear stress ($\unit{N m^{-2}}$).

When the bed is cohesive ($p_{m}~>~p_{m,cr}$), the mud and sand fluxes are proportional to the mud and sand fraction. The erosion rate of mud is calculated by the Partheniades-Krone formulation (\citeauthor{partheniades1965} \citeyear{partheniades1965}; Eq.~\ref{eq:PK1}) similar to the non-cohesive regime. The erosion rate for sand on the other hand was based on the entrainment of mud, because sand particles are included in the cohesive matrix (Eq.~\ref{eq:EE2}). In this way sand can only be eroded when mud is eroded. Bed load transport was assumed to be zero in the cohesive regime.
\begin{equation}
\ E_{s} = E_{m} \label{eq:EE2}
\end{equation}

Sediment suspended following the Partheniades-Krone formulation was further described by the advection-diffusion equation. Sand and mud behave independently in suspension and segregation will occur with low concentrations \citep{torfs1996}. For simplicity we assumed a constant settling velocity of $0.25~\unit{mm /s^{-1}}$ for mud, ignoring that settling velocity depends on flocculation influenced by concentration, residence time, salinity, pH, turbulence and biochemical effects \citep[e.g.,][]{mietta2009}. The settling velocity is typical for fluvial mud \citep[$0.1-0.4~\unit{mm s^{-1}$,][]{temmerman2003}, which we supply in our default run, and is relatively low for marine mud. Deposition of mud is determined by the concentration, settling velocity and the step function similar to Eq.~\ref{eq:PK1}. However, many studies show that deposition is continuous and a threshold for deposition is therefore absent \citep[short review in][]{sanford2008}. This is approached in the model by setting a very high critical shear stress for sedimentation (Table~\ref{sediment}).

Bed level changeare caused by the divergence of sediment fluxes . To track the mud and sand fractions in the bed, a bed module was used with a mixed Eulerian-Lagrangian approach \citep{kessel2011,deltares2012} similar to \citet{hir2011} and \citet{sanford2008}. An active Lagrangian layer of $10~\unit{cm}$ was used where sediment exchange occurs with the water column. This active layer had a constant thickness and moved through the vertical framework with bed aggradation and degradation. Below the active layer we used several vertically fixed Eulerian layers to store bed composition in the vertical (Table~\ref{input}). The advantage of Eulerian bed-layers is that  artificial mixing by vertically moving layers is prevented. The advantage of a  Lagrangian active layer is that the thickness is constant, which is desired because  strong bed armouring is prevented and the thickness affects the time scales of the system \citep{deltares2012}.

To speed up morphodynamic calculations the bed level change in each time step, calculated from the divergence of sediment fluxes, was multiplied with a morphological factor of $400$ (Table~\ref{input}). Extensive studies showed reasonable results up to a morphological factor of $1000$, though it is recommended  not to go above $400$ \citep[Fig.~\ref{morfacs}]{roelvink2006,wegen2008a}. Using a morphological factor is an efficient way of speeding up long-term morphodynamic calculations that is widely used \citep{roelvink2006,wegen2008a,hir2015,dam2016}.

When the water level changes during a tidal cycle, flooding and drying of intertidal area occurs. To prevent complicated and time-consuming hydrodynamic calculations with very small water depths a threshold is set for drying and flooding (Table~\ref{input}). When the water depth is below this threshold the velocity is set to zero. Since the velocity in dry cells is zero, there is no sediment transport in dry cells, even when considerable erosion occurs in a wet cell next to it. Therefore, dry beach and bank erosion was implemented to drive lateral bed lowering. A user-defined factor (Table~\ref{input}) determines the fraction of the erosion flux that is assigned to the adjacent dry cells.

The transverse bed slope effect is a very important parameter for bar dimensions and behaviour in morphological models that is often used as a calibration parameter \citep{schuurman2013}. In estuary models the transverse bed slope effect is often set to be much stronger than the advised default settings to prevent unrealistically steep banks and narrow bars and channels from forming\citep{wegen2012}. The reason for this is unclear but unravelling this is beyond the scope of the present paper so we use settings similar to earlier studies \citep{wegen2012}. We used the transverse bed slope predictor of \citet{koch1980} as extended by \citet{talmon1995}:
\begin{equation}
\ f(\theta) = \alpha \theta^{\beta}
\label{eq:KF}
\end{equation}
where $\theta$ is the shields parameter, $D$ median grain size ($\unit{m}$), $H$ the water depth ($\unit{m}$) and $\alpha=0.2$, much lower than the default of $1.5$ for rivers, and $\beta=0.5$.

\begin{table*}[t]
        \caption{Parameters for processes and numerics}
        \begin{tabular}{ l c c c }
        Parameter & symbol & unit & value \\
        \tophline

Time step & $dt$ & $\unit{min}$ & $0.3$ \\
Spin up time at cold start & $-$ & $\unit{min}$ & $1.44 \times 10^{4}$ \\
Threshold depth drying/flooding & $-$ & $\unit{m}$ & $0.08$ \\
Min water depth for bed level change & $-$ & $\unit{m}$ & $0.05$ \\
Erosion adjacent dry cells & $-$ & $-$ & $0.5$ \\
Morphological factor & $Morfac$ & $-$ & $400$ \\
Transverse bed slope parameter & $\alpha$ & $-$ & $0.2$ \\
Transverse bed slope parameter & $\beta$ & $-$ & $0.5$ \\
Eulerian bed storage layer thickness & $-$ & $\unit{m}$ & $0.1$ \\
Active layer thickness & $-$ & $\unit{m}$ & $0.1$ \\
\bottomhline
\end{tabular}
\label{input}
\end{table*}

The Engelund-Hansen transport formulation was chosen because other relations, in particular \citet{vanrijn1993,vanrijn2004,vanrijn2007}, resulted in higher bars and much deeper and straight channels with sudden (up to $90~^\circ$) sharp bends, which would require transverse bed slope parameters that differ two orders of magnitude from the theoretical value in estuarine settings \citep{wegen2012}. Furthermore, changing bed slope parameters does not fix the channel pattern issues. For long-term morphological modelling Engelund-Hansen produces more realistic morphologies. The disadvantage of our method is that the present code for sand-mud interaction with Engelund-Hansen does not yet incorporate a gradual transition in critical shear stress for erosion between the cohesive and non-cohesive regime. Additionally, mud would ideally erode proportionally with sand in the non-cohesive regime as sand erodes with mud in the cohesive regime, but this is not yet implemented for Engelund-Hansen and is therefore also not described in our method section. These issues are beyond the scope of the present paper and require further research and model code development.

\subsection{Model schematization}
The modelled domain is $30~by~15~\unit{km}$ of which $10~by~15~\unit{km}$ is sea area (Fig.~\ref{initialbathy}). The grid is rectilinear with a  resolution.  that varies between $50~by~80$ and $125~by~230~\unit{m}$. Cell size increases from the initial estuary shape to the sides and offshore to increase resolution in regions of interest and to decrease computation time. The initial bathymetry is in the shape of an idealised funnel-shaped estuary. This exponential shape was also found in previous modelling research \citep{lanzoni2002,canestrelli2008,lanzoni2015} and obtained from field data \citep{savenije2015}. The estuary is $3~\unit{km}$-wide at the mouth and decreases exponentially to a channel of $300~\unit{m}$ wide over $20~\unit{km}$. The bed level linearly increases in elevation from $-2$ at the mouth to $2~\unit{m}$ at the upstream boundary and $2~to~3~\unit{m}$ on dry land \citep{wegen2008a}. The sea has a maximum depth of $15~\unit{m}$. \citet{wegen2008a} argued that initial bathymetry does not greatly affect the dynamic equilibrium shape, because dry-cell or bank erosion is allowed in the model and the model will therefore develop a self-formed (alluvial) estuary shape. However, initial shape affects the time needed to form the equilibrium planform shape as well as the size of the ebb delta in the absence of waves and littoral transport, which is the default situation in our idealised estuary. We therefore started with a funnel shape to save calculation time and decrease the size of the ebb tidal delta. The shape is given as:
\begin{equation}
\ W = W_{mouth} e^{\left(\frac{-x}{L_b}\right)}
\label{eq:eshape}
\end{equation}

% \frac{200}{2}+e^{\left(x\frac{\ln \left(\frac{3000}{\left(2\right)}\right)}{20000}\right)}
% \frac{200}{2}+\left(\frac{3200}{2}-\frac{200}{2}\right)^{\frac{x}{20000}}
% 200+\left(3200-200\right)e^{\left(-\frac{x}{3362.6}\right)}
% 3000e^{\left(-\frac{x}{3362.6}\right)}

where $W_{mouth}=3000~\unit{m}$ is the width of the estuary at the mouth , $L_b=3362.6~\unit{m}$ is the e-folding distance over which the width of an exponential channel is reduced by a factor of $e$, and $x$ is distance from the mouth ($\unit{m}$). The shapes of modelled estuaries are characterised by the funnel shape parameter \citep{davies2010} calculated as e-folding length normalised by mouth width at that point in time (Eq.~\ref{eq:fs}). Lower values of the characteristic funnel length indicate stronger funnelling in the sense of more rapid narrowing from the mouth in landward direction. In this way estuary shape is normalised by estuary size.
\begin{equation}
\ S_b = L_b/W_{mouth}
\label{eq:fs}
\end{equation}

\begin{figure*}[t]
\centering
\includegraphics[width=12cm]{initialbathy_width12cm_mdlnr1}
\caption{Initial bathymetry with model boundaries and cross-section (red line) for analysis. Initial depth increases linearly upstream and width decreases exponentially (Eq.~\ref{eq:eshape}). Coordinates are defined at the coastline with the channel centreline and mean sea level (MSL) as origin.}
\label{initialbathy}
\end{figure*}

[revised manuscript text omitted]

\begin{figure}[t]
\includegraphics[width=8.3cm]{peak_u_ratio_54_109}
\caption{Flood/ebb peak velocity ratio over time for the first $5~\unit{km}$ of the default estuary (run~01) integrated over the total area (black~squares), area above mudflats (brown~circles) and areas above (cyan~triangles) and below (blue~triangles) $-1~\unit{m}$ bed level.}
\label{peak}
\end{figure}

\subsection{Effects of mud supply}
We will now compare other scenarios (run 03, 10 and 09) with the default run (01). In most scenarios mud is accumulating on the flanks of the estuary where the velocities are low and in the upper estuary where it covers a relatively large fraction of the width (Fig.~\ref{mud}f,~g~and~h). Locally, mud accretes on bars that are rather stable (e.g. Fig.~\ref{mud}g~and~h, on the ebb delta). The initiation of mud flats proceeds by the positive feedback identified in the model description: once mud starts settling somewhere, the mud fraction in the bed rapidly increases beyond the critical mud fraction for mud-dominated behaviour. As a consequence, the critical shear stress for sand erosion equals the entrainment threshold of mud (Eq.~\ref{eq:EE2}). The mud-dominated mixed sediment thus becomes more difficult to erode and more rapid aggradation of mud is likely to occur.

Migration rates of channels decrease considerably due to the addition of cohesive material (Fig.~\ref{mud_time}a--h). Bar splitting and merging related to chute cut-offs and avulsion also reduce with increasing mud concentrations. In Fig.~\ref{mud_time}a—-dh channels move through a cross-section at the mouth through time, though slower for a larger mud supply. For example, a large bar forms in the mouth after about $1100~\unit{yr}$ in the scenario with only sand (Fig.~\ref{mud_time}a) and in the scenario with a mud supply of $50~\unit{mg L^{-1}}$ (Fig.~\ref{mud_time}d). In the run with mud, the bar is covered with mud and becomes fixed while the large bar in the scenario with only sand migrates about $1~\unit{km}$. Absolute bed level changes also indicate that dynamics are decreased with mud input (Fig.~\ref{sourcemud_time}y--II), because there is less bed level change per timestep.

The mudflats have a strong effect on the final shape of the estuary (Fig.~\ref{mud}). Firstly, an increase in fluvial mud input concentration leads to stronger self-confinement of the estuary. By depositing mud on the sides of the estuary, the banks become more stable and limit (further) erosion due to an increased critical shear stress. Self-confinement of estuaries is clearly observed when the models with mud supply are compared to the control run without mud (Fig.~\ref{mud}a). The runs with mud are narrower and have a smaller surface area due to filling of the initial bathymetry, while the sand run has expanded in size. Consequently, the braiding index lowers with increasing mud concentration (Fig.~\ref{mud_length}e--h). In contrast, estuarine surface area continues to increase over time for the control run with only sand (Fig.~\ref{mud_time}q). After the initial rapid change the increase in area and width is linear, driven by dynamic channels and bars and is unhindered by bank stability. This suggests that there is no equilibrium shape under these conditions as is also reflected in the absolute and net bed level change (Fig.~\ref{mud_time}y-VI~and~III). The absolute bed level change does not approach a constant value and the net bed level remains negative, demonstrating the sand-only estuary to be a continuously exporting system.

\begin{figure*}[t]
\includegraphics[width=17cm]{maps_width17cm_runmud_t582}
\caption{Effects of mud supply concentration (run 03, 10, 01 and 09). Left column shows final bathymetry of model runs after $2000~\unit{yr}$ and the right column shows mud fractions in the top layer of the bed. Runs with (a,e) $0~\unit{mg L^{-1}}$, (b,f) $5~\unit{mg L^{-1}}$, (c,g) $20~\unit{mg L^{-1}}$ (default) and (d,h) $50~\unit{mg L^{-1}}$ fluvial mud supply concentration.}
\label{mud}
\end{figure*}

For estuaries with fluvial mud, higher concentrations lead to narrower (Fig.~\ref{mud_length}i--l and Fig.~\ref{mud_time}i--I) and smaller (Fig.~\ref{mud_time}q--t and Fig.~\ref{hypso}) estuaries. Moreover, in some places the width of the estuaries with mud supply is narrower than the initial width, supporting our finding that the initial bathymetry is of limited influence because the system is able to fill and to expand (see methods). Furthermore, tidal bars become higher with increasing mud concentrations, which results in an increased average bed level (Fig.~\ref{mud_length}a--d). Furthermore, mMud is almost nowhere deposited in the channels and does therefore not limit bed erosion by cohesion (Fig.~\ref{hypso}). We thereforeAs a result we infer that the shallower channels in increasingly muddy estuaries mainly result from the decrease in estuary width and concurrent reduction of intertidal area, tidal range and tidal currents (Fig.~\ref{mud_length}).

\begin{figure}[t]
\includegraphics[width=8cm]{hypso_cum_width8cm_run2}

[revised manuscript text omitted]

\begin{figure*}[t]
\includegraphics[width=17cm]{concl_width17cm}
\caption{Most important large-scale morphological parameters after $2000~\unit{yr}$ as a function of the varied boundary conditions: fluvial mud supply concentration , tidal range  and fluvial discharge . (a-c) funnel-shape parameter, (d-f) mouth with (in blue colours) and mud flat width (brown colours) at the mouth  and (g-i) total area (blue colours) and mud covered area (brown colours $\unit{km^2}$). Data indicated in light blue and light brown use a more aggressive masking technique in which high mud flats (higher than $0.5~\unit{m}$ below high water level) are masked from the estuary shape from which area,

width and funnel-shape are calculated. Light grey areas indicate models in the transition from estuary to delta. Dark grey indicate models that evolved into a delta.}
\label{concl}
\end{figure*}

\subsection{Comparison to real estuaries}
Model conditions fall within the parameter space of natural estuaries \citep[Fig.~\ref{prandlefig}; Table~\ref{prandletab};][]{prandle2005,leuven2016}. The model has typically larger discharges than the small UK estuaries, but discharge and tidal amplitude falls well within the range of estuaries worldwide.

Several aspects of the bar patterns are further indications that the numerical models reproduce important emergent phenomena of real estuaries. For example, we observe ebb- and flood-dominated channels that are unique for tidal systems \citep{veen1950,ahnert1960}. Typical bar dimensions obtained from the models are in good agreement with natural estuaries from a large dataset \citep{leuven2016}; for example tidal bar length is approximately 7 times the partitioned bar width (maximum bar width devided by barb channels). Furthermore, bar length approximates local width of the estuary. Bars without mud are generally longer and wider for this model study and relative to the local estuary width. Bars in models are also slightly bigger with marine mud supply rather than for fluvial mud supply. The braiding index is strongly related to estuary width as found for natural estuaries \citep{leuven2016} and in agreement with the relation between tendencies to form floodplain in rivers and the resulting relation between channel aspect ratio and bar pattern \citep{kleinhans2010, kleinhans2011, schuurman2016}.

\begin{figure*}[t]
\includegraphics[width=17cm]{bar_length_to_width_and_dependence_width}
\caption{(a) Bar length versus partitioned bar width and (b) bar length against local estuary width. Model results plot in the same range as the data of the natural estuaries \citep{leuven2016}.}
\label{bar_length_to_width_and_dependence_width}
\end{figure*}

The completed model runs show mud flat characteristics and behaviour broadly comparable to natural estuaries. Spatial Ttrends in the field data, shown earlier (Fig.~\ref{datafigure}), generally agree well with the model results. We observe similar depositional areas of mud on the sides of the estuaries in the form of mudflats (Fig.~\ref{datafigure}a--e and Fig.~\ref{mud}e--h). In the centre of the lower estuary there is little mud compared to the mudflats on the sides. However, some mud is observed on some of the bars in the Western Scheldt (e.g.~Fig.~\ref{datafigure}c) as in some model scenarios (Fig.~\ref{mud}h). Comparison of the observed and modelled hypsometries (Fig.~\ref{hypso}~and Fig.~\ref{datafigure}g~and~h) shows that mud is deposited at comparable elevations, mostly at intertidal areas and more specifically namely around mean water level. We observe a strong increase in mud flats with the strongest increase is cumulative area. We observe similar depositional areas of mud on the sides of the estuaries in the form of mudflats (Fig.~\ref{datafigure}a--e and Fig.~\ref{mud}e--h). In the centre of the lower estuary there is little mud compared to the mudflats on the sides. However, some mud is observed on some of the bars in the Western Scheldt (e.g.~Fig.~\ref{datafigure}c) as in some model scenarios (Fig.~\ref{mud}h).

The fluvial mud scenarios have relatively large fractions of width covered by mud flats in the upper estuary as in the single-channel upper estuaries in the Netherlands. Indeed, most mud is deposited in the middle and upper estuary, where the estuary consists of only one channel. This is also clearly observed in the McLaren \citet{mclaren1993,mclaren1994} dataset of the Western Scheldt (Fig.~\ref{datafigure}a). The tidal river was found to contain more mudflats than the lower estuary

(Fig.~\ref{datafigure}f). Note that Fig.~\ref{mud_length} underestimates modelled mud flat surface shown in Fig.~\ref{mud} because many cells are inactive in the computation because they increased in elevation.

Typically in the model, marine mud does not settle much and far in the estuary. This is not what is observed in the Western Scheldt. \citet{verlaan2000} studied the marine versus fluvial distribution of mud through the estuary. He found a sharp increase in mud fraction in the bed between Lillo and Saeftinge from $10~\unit{\%}$ to $75~\unit{\%}$, which is far upstream in the narrow single channel system. This might be a consequence of the assumption that settling velocities for fluvial and marine mud are the same while typically settling velocities of marine mud are significantly higher due to flocculation. Marine macroflocs settling rates might be as high as a few $\unit{mm s^{-1}}$ \citep{mietta2009,leussen2011}. It is also a likely possibility that the Western Scheldt is not comparable to our modelled system considering marine mud deposits, because the salinity intrusion of the Dovey and Western Scheldt is incomparable. Mud deposition data from the Dovey estuary is unavailable although mud flats and muddy marshes are easily observable on aerial imagery \citep{leuven2017}.

In the model we observe sharp transitions between areas without mud in the bed ($<10~\unit{\%}$) and areas with very high mud fractions ($70\text{-}100~\unit{\%}$). This is also observed in the Western Scheldt according to \citet{ledden2002}. More gradual transitions of mud are expected for $w_s \times c / M >> 1$, where $w_s$ is fall velocity, $c$ is concentration and $M$ is the erosion parameter \citep{ledden2002}. All our models have ratios below 1 in agreement with conditions in the Western Scheldt and probably in agreement with conditions in the Dovey given the clearly observable sand-mud transitions on imagery.

In the Western Scheldt the fluvial mud supply varies between $100~\times 10^{6}$~and~$300~\times 10^{6}~\unit{~Gg~ kg yr^{-1}}$ at the Rupple mouth \citep{traverniers2000}. In the model the mud input is $63~\times 10^{6}~\unit{~Gg~ kg yr^{-1}}$. The mean discharge of the Scheldt, about $120~\unit{m^3 s^{-1}}$, is about $20~\unit{\%}$ higher than the default model scenario, while the sediment input is at least $60~\unit{\%}$ higher. This higher mud load might explain why the Western Scheldt has more mud deposits. In the field case this occurs more on bars than on the sides compared to the models, which difference we may partly attribute to the embankment and limited space to form mud flats and partly attribute to spatially and temporally varying mud characteristics in the Western Scheldt.

The default scenario shows that the velocity amplitudes are flood-dominant in shallow areas and ebb-dominant in the channels (Fig.~\ref{hydro}h). This is in general agreement with most earlier findings about tidal asymmetry \citep[e.g.][]{speer1985,friedrichs1988,wang2002,moore2009,} including model studies on the Dovey \citep{robins2010,brown2009,brown2010}. Our findings generalise these earlier trends, because our estuary is self-formed, while several bathymetries tested in previous research are strongly simplified or arbitrary chosen and might not represent a realistic state of an estuary \citep{speer1985, , meaning that flood- or ebb dominance could be the result of the imposed combination of initial condition and boundary conditions. In contrast with our results, these case-studies found higher flood-peak velocities upstream \citep{brown2010,robins2010}. This is attributed to more intertidal area upstream that promotes flood-dominance \citet{moore2009,brown2010,robins2010}. Our default model showed shows stronger ebb-dominant peak velocities in the landward part (Fig.~\ref{hydro}g), which is caused by a higher river discharge in our model which causes ebb-asymmetry.

Over time, our model evolved from a net exporting system to a dynamic equilibrium with balanced import and export (Fig.~\ref{finalbathy_5}e and j). As more intertidal area and mudflats formed in the estuary, these areas gradually transformed from ebb to flood-dominant peak velocities (Fig.~\ref{peak}). Especially the mudflats show much stronger flood-dominant peak velocities and a faster change over time than the intertidal area in general. This is because mudflats are significantly higher and have an elevation near high water level, while typical sandy shoals only have a maximum height between low and mean water level. This matches well with the sediment budget of the model that shows net import of sediment resulting from mud import and sand export. This trend is also observed, most likely for the same reason, in the Western Scheldt on the basis of separate sand and mud balances \citep{cleveringa2013}. Mud trapping is very efficient as the import is significant even though the duration asymmetry and peak velocity asymmetry are ebb-dominant in most of the estuary. This, again, shows that the spatial variation in ebb and flood asymmetry is very important for understanding if the estuary will grow or fill. Moreover, representation of tidal asymmetry by width-averaged velocity ratios are insufficient and misleading in the presence of significant mud deposits. Due to mud deposition, the elevation of intertidal flats increases, which is therefore essential to change an estuary from exporting to importing or towards an equilibrium system.

\subsection{Transition from estuary to delta}
The parameter space of \citet{prandle2005} suggests that tides and river flow are sufficient conditions to explain the bathymetry of an estuary, with longer tidal reaches with larger river inflow (Fig.~\ref{prandlefig}). This trend is not reproduced in the idealised model scenarios that typically have a tidal reach of $5-15~\unit{km}$ long, but plot far above the line of $20~\unit{km}$ in Fig.~\ref{prandlefig}. Likewise, the trend is not clear in the dataset either \citep[Fig.~3 in][]{prandle2005}. Rather, we observe the opposite trend: shorter estuaries, or even deltas, form with larger river discharges and longer estuaries form in higher tidal ranges. Possibly longer estuaries form for larger total flow from the combination of tide and river. We found much stronger effects of mud supply, suggesting that the tide-discharge parameter space needs to be extended with sediment supply.

\begin{figure}[t]
\includegraphics[width=8.3cm]{prandle}
\caption{Tidal amplitude plotted against river discharge for real world estuaries and modelled scenarios. Field data is used from \citet{prandle2005} for estuaries in the UK and several other sources for different estuaries over the world. Lines indicate estimations of estuarine length by \citet{prandle2005} of $5, 10$ and $20~\unit{km}$ from left to right.}
\label{prandlefig}
\end{figure}

\begin{table*}[t]
\caption{Ranges of conditions in mixed estuaries at temperate zones \citep{prandle2005} compared to
values for the modelling results.}
\begin{tabular}{ l c c c }
Parameter & Unit & Range & Model \\
        \tophline
        Tidal amplitude & $\unit{m}$ & $1-4$ & $1.5$ \\
        Velocity amplitude & $\unit{m s^{-1}}$ &
        $0.5-1.25$ & $0.5-1$ \\
        River discharge & $\unit{m^{3} s^{-1}}$ & $0.25-3000$ & $100$ \\
        Associated current & $\unit{m s^{-1}}$ &
        $ 0.001-0.01$ & $??$ \\

        Depth at the mouth & $\unit{m}$ & $1-20$ & $2$ \\
        Tidal intrusion length & $\unit{km}$ & $2.5-100$ & $~15$ \\
        Age & $\unit{yr}$ & $100-15000$ & $2000$ \\
        Fall velocity & $\unit{mm s^{-1}}$ & $0.5-5$
        & $0.25 (mud), 41 (sand)$ \\
        \bottomhline
        \end{tabular}
        \label{prandletab}
\end{table*}

As our model runs cover transgressive and regressive trends as effects of tides, river, waves and sediment supply on morphology we attempted to position our results in the traditional ternary classification diagrams for deltas of \citet{galloway1975}. An expanded version of this classification system includes all coastal environments, where larger river influence leads to delta development and low or absence  of river influence leads to lagoons, strandplains and tidal flats \citep{dalrymple1992,boyd1992}. Qualitatively our results also show that for higher river discharge the estuarine system transitions to a deltaic system (Fig.~\ref{concl}c--i) by filling of the estuary. Note that the width did not decrease because a small tidal basin north of the river mouth affected the automated calculation of the width of the system (Fig.~\ref{Q}a). We also observed a transition to deltas when the tidal range was decreased (Fig.~\ref{concl}b--h), so that the relative power of the river increases in qualitative agreement with the classification diagram.

However, the most important findings of our research are more difficult to relate to these diagrams. We found that an increase in mud supply concentration leads to confining and filling of the initial estuary shape (Fig.~\ref{concl}a--g) leading to a decrease in total area and width at the mouth, while the mud covered area and mud flat width at the mouth increased and is more delta-like. \citet{orton1993} found that smaller grain size leads to narrower channels in deltas and a tendency to avulse rather than have migrating channels. We observe similar behaviour in the model scenarios but here this is related not merely to grain size but to the supply rate.

Alternatively, \citet{dalrymple1992} and \citet{ boyd1992} developed a classification system with a fourth dimension based on the evolution of coastal systems by defining it as a prograding or transgressive system on the basis of sea level rise and sediment supply. This system disregards the possibility of an equilibrium without progradation and without transgression through combinations of sediment supply but otherwise similar hydrodynamic conditions. Our models with different fluvial mud supply concentrations lead to distinct different morphologies but would plot on the same coordinates in these diagrams. Additionally, sea level rise is an ambiguous and qualitative variable in their conceptual figure, because it affects the hydrodynamic conditions of the primary ternary diagram. To conclude, our model results for estuaries qualitatively fit in the ternary plots of \citet{dalrymple1992} and \citet{ boyd1992} for deltas when sea level rise is ignored and sediment supply is considered the only variable on the fourth axis.

\subsection{Large-scale equilibrium of estuary shape and dimensions}
Estuaries with fluvial mud supply evolve to large-scale morphodynamic equilibrium (where absolute bathymetry change is constant, Fig~\ref{finalbathy_5}c, net bathymetry change is zero, Fig~\ref{finalbathy_5}d, and net export equals import, Fig~\ref{hydro}d) with dynamic channels and bars, but in the absence of mud keep expanding continuously by bank erosion due to channel migration. This agrees with the continuously exporting estuaries in the numerical models of \citet{wegen2008b} and with the physical experiments of \citet{kleinhans2015} with perpetually expanding tidal basins in cohesionless sand. After a rapid adjustment of basin size and bar and

channel pattern the experiments developed to near-equilibrium but never attained equality of sediment import and export. Our scenario without discharge is similar to these experiments and shows the same evolution, including the rapid adjustment and continuous erosion in a low-dynamic state (Fig.~\ref{Q_time}d--VI). In braided rivers such unhindered bank erosion leads to a 'threshold channel' \citep{parker1978} with an equilibrium width related to the upstream flow discharge and the threshold for sediment motion. This theory was earlier suggested to be valid for tidal basins \citep{kleinhans2015}. However, unlike rivers, estuaries are not limited by discharge because tidal prism can continue to increase as the estuary enlarges, leading to a potentially positive feedback only limited by friction. In other words, estuaries may expand to much larger systems because the tidal prism adapts to estuary size and flow velocities and entrainment rates will not decrease with basin size unless opposed by cohesion. This proved to be the case in our models with mud. From this we conclude that development to an equilibrium shape for estuaries requires some form of apparent cohesion from mud, from species with sediment-binding effects and from unerodible valley walls.

This explains why previous studies found large-scale equilibrium in estuaries: these imposed a fixed estuary shape and size in 1D simulations \citep[e.g.,][]{lanzoni2002,schuttelaars2000,todeschini2008} or imposed non-erodible boundaries in 2DH \citep[e.g.,][]{hibma2003,wegen2008a}.

[revised manuscript text omitted]

\begin{figure*}[t]
\includegraphics[width=17cm]{maps_width17cm_runtide_t582}
\caption{Effects of tidal range (run 06, 01, 05 and 20). Left column shows final bathymetry of model runs after $2000~\unit{yr}$ and the right column shows mud fractions in the top layer of the bed. Run with (a,e) $4~\unit{m}$, (b,f) $3~\unit{m}$ (default), (c,g) $2~\unit{m}$ and (d,h) $1~\unit{m}$ tidal range.}
\label{tide}
\end{figure*}

\begin{figure*}[t]
\includegraphics[width=17cm]{length_width17cm_run3}
\caption{Hydrodynamics and morphology along estuaries with different tidal ranges after $2000~\unit{yr}$. From left to right column: model with $4$ (06),~$3$ (default, 01),~$2$ (05)~and~$1~\unit{m}$ (20) tidal range. (a--d)~Minimum, mean and maximum bed elevation and, high and low water level and minimum  and maximum initial bed level, (e--h)~braiding index, (i--l)~estuary width defined as: the initial width, maximum reach over the whole scenario run, the width of wet cells in the model, width defined by a threshold value that is used to mask the cells that are around the dry-wet cell threshold. (m--p)~intertidal area and mud cover as percentage of the total area, (q--t) tidal range and (u--x) peak ebb and flood velocities.}
\label{tide_length}
\end{figure*}

\begin{figure*}[t]
\includegraphics[width=17cm]{overtime_width17cm_run3}
\caption{Hydrodynamics and morphodynamics over time for estuaries with different tidal ranges. From left to right column: model with $4$ (06),~$3$ (default, 01),~$2$ (05)~and~$1~\unit{m}$ (20) tidal range. (a--d)~Bathymetry of the cross-section at the mouth plotted over time, (e--h)~mud

fraction in top layer of cross-section at the mouth, (i--l)~estuary width at
$1$,$~4~$and~$8~\unit{km}$ from the mouth, (m--p)~funnel-shape parameter, (q--t)~estuarine
surface area, (u--x)~intertidal area and mud in the bed relative to the total area, (y--II)~absolute bed
level change and (III--VI)~net bed level change.}
\label{tide_time}
\end{figure*}
\clearpage

\begin{figure*}[t]
\includegraphics[width=17cm]{maps_width17cm_runwave_t329}
\caption{Effects of mud source in the presence of waves (run 28, 27, 25 and 29). Left column shows
final bathymetry of model runs after $1250~\unit{yr}$ and the right column shows mud fractions in
the top layer of the bed. Run with (a,e)~only sand, (b,f)~marine mud input (default), (c,g)~marine
and fluvial mud input and (d,h)~fluvial mud input.}
\label{wave}
\end{figure*}

\begin{figure*}[t]
\includegraphics[width=17cm]{length_width17cm_run5}
\caption{Hydrodynamics and morphology along estuaries for different mud sources in the presence
of waves after $2000~\unit{yr}$. From left to right column: model with only sand (28), marine mud
supply (27), supply from both boundaries (25) and fluvial supply (29)(default). (a--d)~Minimum,
mean and maximum bed elevation and, high and low water level and minimum and maximum initial
bed level, (e--h)~braiding index, (i--l)~estuary width defined as: the initial width, maximum reach
over the whole scenario run, the width of wet cells in the model, width defined by a threshold value
that is used to mask the cells that are around the dry-wet cell threshold. (m--p)~intertidal area and
mud cover as percentage of the total area, (q--t) tidal range and (u--x) peak ebb and flood velocities.}
\label{wave_length}
\end{figure*}

\begin{figure*}[t]
\includegraphics[width=16cm]{overtime_width17cm_run5}
\caption{Hydrodynamics and morphodynamics over time for estuaries for different mud sources in
the presence of waves. From left to right column: model with only sand (28), marine mud supply
(27), supply from both boundaries (25) and fluvial supply (29default). (a--d)~Bathymetry of the cross-
section at the mouth plotted over time, (e--h)~mud fraction in top layer of cross-section at the
mouth, (i--l)~estuary width at $1$,~$4$~and~$8~\unit{km}$ from the mouth, (m--p)~funnel-shape
parameter, (q--t)~estuarine surface area, (u--x)~intertidal area and mud in the bed relative to the
total area, (y--II)~absolute bed level change and (III--VI)~net bed level change.}
\label{wave_time}
\end{figure*}
\clearpage

\begin{figure*}[t]
\includegraphics[width=17cm]{morfacs_width17cm}
\caption{Resulting bathymetries from runs with different morphological acceleration factors of (a)
10, (b--c) 100 and (d--e) 400 after (a,b,d) 50 and (c,e) 500 years.}
\label{morfacs}
\end{figure*}

\authorcontribution{The authors contributed in the following proportions to concept and design, modelling, analysis and conclusions, and manuscript preparation: LB(40,100,50,70~\%), TK(10,0,0,10~\%), JL(5,0,5,0~\%) and MK(45,0,45,20~\%).}

\competinginterests{The authors declare that they have no conflict of interest.}

%\disclaimer{?}

\begin{acknowledgements}
This work is part of the PhD project of LB in the project 'Turning the Tide' funded by the Dutch Technology Foundation (STWTTW) of the Netherlands Organisation for Scientific Research (NWO), Vici-grant 016.140.316/13710 to MK. We would like to thank Pierre Weill and the anonymous reviewer for their contributions to improve the manuscript. Additionally, we are grateful to Mark Macklin, Paul Brewer, Jaco Baas and Alan Davies for a field site visit and discussions on the Dovey estuary and Marco Schrijver of Rijkswaterstaat for field visits and discussion on the Western Scheldt. We acknowledge Bert Jagers of Deltares for help with the sand-mud interaction module of Delft3D, Mick van der Wegen for discussions on long-term modelling, MSc thesis student Samor Wongsoredjo for testing boundary conditions for waves and Anne Baar for comments on an earlier version and discussion. Reviewers will be acknowledged.
\end{acknowledgements}

\bibliographystyle{copernicus}
\bibliography{paper1}

\end{document}

---

## Author Response (AR2)

Dear editor,

We would like to thank referee #2 for his/her second review, which helped us to clarify the manuscript. We have now addressed all of the reviewer's comments. Please find below a point by point response of how we implemented the suggestions in the manuscript.

On behalf of all authors,
Lisanne Braat
* * *
*Review of referee #2:*

*The authors made some efforts to improve their manuscript, but also only partly responded or even ignored some of my previous comments. I will therefore reformulate the problems that would need to be addressed before the paper can be definitely accepted:*

*-The representation of short waves in the model is still not clearly explained in section 2.1. According to the elements provided by the authors, it seems that the only effects that is represented is the enhanced bed shear stress due to orbital wave motions. If this is the case, this should be clearly stated around L.21-28. Then, does Tau_cw of eq. (5) stands for "currents & waves"? How is it computed? Does it apply to the circulation model and the sediment transport model or only the second one?*

Indeed, the methodology lacked an explanation of the waves used in the model. We now detail this in 2.1 (P6-L11): "The SWAN module was used to implement the effect of short waves. We used two-way coupling between the flow and wave module with an interval of $3~\unit{h}$. At four stages during every tidal cycle the wave conditions were calculated in SWAN from the current situation of the morphological model. The waves enhanced turbulence and bed shear stress by wave-driven currents in the morphological model. The sediment transport was only affected by the enhanced bed shear stress by wave-current interaction and not by enhanced turbulence. "
Regarding the effect of the waves on sediment transport, we now state that the enhanced bed shear stress is the only effect of the waves. Additionally we clarified that Tau_cw means the maximum bed shear stress due to currents and waves below the Eq. 5 (P7-L21) and clarify that the enhanced bed shear stress affects the mud transport directly (P11-L12) and the sand transport only indirectly by the effect of the bed shear stress on the currents (P11-L15).

*-Tidal boundary conditions. Unlike along the coastlines of the Netherlands, most coastal zones around the world are exposed to tides propagating mainly shore-normal. Why applying an alongshore phase lag to M2 and not a cross-shore one? Why not prescribing tides along the Western boundary?*

We chose an alongshore tide because we used the Dovey Estuary as inspiration, where the tidal propagation is also alongshore. We added this in the text (P10-L20). Most of our data is from the UK and the Netherlands, so we did not see this as a problem.
Although many estuaries are exposed to shore-normal tides, we do not expect that the orientation of the tide will make a large difference to our results since the phase difference between the cross-shore borders is very small. We would expect that only the mud deposits along the coast would be spread more symmetrically.
We did not prescribe tides along the Western boundary because the third open boundary created instabilities in the velocity in the corners of the model. The chosen tide is exactly cross-shore and not oblique, therefore the closed Western boundary does not affect the tide. This is now included in the manuscript (P10-L21).

*-The section 4.3 of the discussion totally misses the effects of short waves. As explained in my previous review, an oceanic basin large-enough to have meso-tidal ranges has usually substantial short waves as well. As explained in nowadays numerous published studies (e.g. Bertin et al., 2009; Nahon et al., 2012; Wargula et al., 2014; etc...), short waves drive a range of processes that promote flood-dominance and tend to counteract the ebb dominance that estuaries and inlets develop due to tidal asymmetry. In the absence of short waves, this is not surprising that the authors model predicts continuous enlargement of the estuary mouth, at least without mud. Although the studies listed above rather concern tidal inlets and small estuaries, the processes explained are generic and apply to large estuaries and therefore should be included in the discussion.*

We added a section on this subject in the manuscript: "Even though the tidal asymmetries in the model are comparable to many estuaries, waves are largely simplified. Waves are known to promote flood-dominance by different processes such as wave breaking (Bertin et al., 2009; Nahon et al., 2012; Wargula et al., 2014). We expect that the inclusion of more wave processes on sediment transport would lead to faster development towards equilibrium by stimulating flood directed transport. If the waves are very strong, we expect filling of the estuary by generation of a spit and the estuary might never have been ebb-dominant in the first place. However, in the absence of waves, the continuous enlargement of estuaries with only sand might be as expected."
Because one of the major effects of waves is the influence on tidal asymmetry, as the reviewer points out, we added this to 4.1 where we discuss ebb-flood dominance.

*Other along the text minor comments*
*-P5, L5: why "Dyfi, i.e. Dovey"? Then estuary should start with a capital letter.*
"estuary" was changed to "Estuary". The two names are mentioned because some publications use the Welsh and others the English name.

*-P5, L6: "...boundary conditions to address the main question:..."*
Replaced "for" with "to address".

*-P6, L11: "caused mainly"*
Switched "caused" and "mainly".

*-P7, L31: erosion rather than degradation?*
Replaced "degradation" with "erosion".

*-P8, L4-8: -P8, L4-15: according to my understanding of Delft3D, bed level changes originate from the horizontal divergence of bedload transport as computed through the Exner equation and erosion/deposition computed as the bottom boundary condition in the suspension transport model. Does the morfac apply to both? Please clarify.*
Yes, the morfac applies to both means of transportation. This sentence was incomplete and now reads: "To speed up morphodynamic calculations the bed level change in each time step, calculated from the divergence of sediment fluxes and the erosion-deposition difference for suspended sediment, was multiplied with a morphological factor of 400 (Table 2)."

*-P8, L26: "differ by two orders..."*
Added "by".

*-P10, L16-17: if waves impact the bed shear stress in the circulation model, then the current velocities will be impacted and sand fluxes computed by the EH formula as well. Please better state that the EH formula does not represent sediment stirring by short waves.*

We clarified the sentence and now reads: "Due to the choice for Engelund-Hansen as sediment transport formulation, sand stirring is excluded. Only indirect sand transport effects occur because the enhanced bed shear stress influences the currents."

*-P11, L5-6: as already commented in my previous review, why not using a parallel version of Delft3D, parallel computing is nowadays totally democratized.*

Parallel computation was not necessary in our study, because we always did a large number of model runs at the same time (as many as the cores available). We believe that computing these runs at the same time on single cores is more efficient than using parallel computation one by one. However we did not check this thoroughly. The computation method should not influence the results. Therefore, now we mention how we computed the results ("Multiple Scenarios were computed at the same time, so parallel computing was not necessary"), but abstain from any further discussion on this method in the manuscript.

*-P15, L4: you may think that I'm playing with words but, if the elevation and the velocity signals are out of phase by pi/2, this is well a phase lag. I keep thinking that you mean that this phase lag is constant along the estuary.*

We are not sure if we understand the reviewer's comment well, but we now clarified what we meant in the manuscript following the explanation below.

Indeed there is a phase difference of pi/2 between the water level and velocity signal. However, for tidal systems we believe that phase lag is the difference between slack water and high or low water as according to Savenije (2016). We want to stress here that there is no lag or delay between the water level and velocity signal. It might be that the definition of phase lag for short waves is different than for tides.

Secondly we also wanted to state that the phase lag is constant along the estuary as the reviewer mentioned, we clarified this in the sentence: "There is no phase lag anywhere along the estuary between water level and velocity, since slack water occurs exactly at high and low water."

[Figure]

Figure 2.6: A wave of the mixed type showing the phase lag between HW and HWS, and between LW and LWS.

*-P15, L27: coma after notably as elsewhere in the manuscript starts by an adverb.*

Corrected.

*-P16, caption figure 4: how do you compute the tidal discharge, you remove the freshwater discharged prescribed at the river boundary?*

The freshwater flux is included in the plot, it shows total discharge. "tidal" is removed from the manuscript in this context.

*-P20, L3: "irrelevant" is too strong, I would suggest "of limited impact".*
Implemented.

*-P24, L26: I would rather say "the effects of short waves on the hydro-sedimentary dynamics of the estuary is limited to the stirring of sediments".*
"only to estuaries with limited wave transport processes are included" was changed to "the effect of short waves on the sediment dynamics is limited to the stirring of sediment".

*-P25, L10: coma after "with this in mind".*
Corrected.

*References:*
*-Bertin X., Fortunato, A.B. and Oliveira A., 2009. A modeling-based analysis of processes driving wave-dominated inlets. Continental Shelf Research 29, 819-834.*
*-Nahon, A., Bertin, X., Fortunato, A.B. and Oliveira, A., 2012. Process-based 2DH morphodynamic modeling of tidal inlets: a comparison with empirical classifications and theories. Marine Geology 291–294, 1-11.*
*-Wargula, A., Raubenheimer, B., Elgar, S., 2014. Wave-driven along-channel subtidal flows in a well-mixed ocean inlet. Journal of Geophysical Research: Oceans, 119 (5), pp. 2987-3001.*
- Savenije, H. H. (2006). Salinity and tides in alluvial estuaries. Elsevier. p36.

**Track changes document compared to the previous manuscript version.**

%% Copernicus Publications Manuscript Preparation Template for LaTeX Submissions
\documentclass[esurf, manuscript]{copernicus}
\begin{document}

\title{Effects of mud supply on large-scale estuary morphology and development over centuries to millennia}

\Author[1]{Lisanne}{Braat}
\Author[2]{Thijs}{van Kessel}
\Author[1]{Jasper R.F.W.}{Leuven}
\Author[1]{Maarten G.}{Kleinhans}

\affil[1]{Utrecht University, Heidelberglaan 2, 3584 CS Utrecht, the Netherlands}
\affil[2]{Deltares, Boussinesqweg 1, 2629 HV Delft, the Netherlands}

\runningtitle{Effects of mud supply on large-scale estuarine morphology}
\runningauthor{Lisanne Braat}
\correspondence{Lisanne Braat (L.Braat@uu.nl)}

\received{}
\pubdiscuss{}
\revised{}
\accepted{}
\published{}

\firstpage{1}

\maketitle

\begin{abstract}

[revised manuscript text omitted]
 ($\unit{m s^{-1}}$), $g$~is gravitational acceleration ($\unit{m s^{-2}}$), $C$~is the Chezy friction parameter ($\unit{m^{0.5} s^{-1}}$) and $v_{w}$~is the eddy viscosity ($\unit{m^{2} s^{-1}}$).

The SWAN module was used to implement the effect of short waves. We used two-way coupling between the flow and wave module with an interval of $3$~$\unit{h}$. At four stages during every tidal cycle the wave conditions were calculated in SWAN from the current situation of the morphological model. The waves enhanced turbulence and bed shear stress by wave-driven currents in the morphological model. The sediment transport was only affected by the enhanced bed shear stress by wave-current interaction and not by enhanced turbulence.

A module was recently developed for mixed sediments which incorporates the effect of bed composition on erosional behaviour and hence morphology \citep{kessel2011,deltares2012}. This module is a partial implementation of \citet{ledden2001b} and \citet{jacobs2011} and tracks spatial and temporal bed composition for multiple grain sizes of sand and mud with erosional characteristics depending on bed composition. In this paper we only used one sand fraction and one mud fraction (Table~\ref{sediment}) and  applied a uniform roughness.

\begin{table*}[t]
\caption{Sediment characteristics applied in the default model. Variation in settling velocity will later be discussed as one of the sensitivity parameters.}
\begin{tabular}{r l c l}
Sediment property & symbol & value & unit \\

\tophline
        Sand \\
        \middlehline
        Settling velocity & $w_{s}$ & $4.4 \times 10^{-2}$ & $m/s$ \\
Median grain size & $D_{50}$ & $3 \times 10^{-4}$ & $m$ \\
        Specific density & $\rho_{s}$ & $2650$ & $kg/m^3$ \\
        Dry bed density & $\rho_{dry}$ & $1600$ & $kg/m^3$ \\
        \middlehline
        Mud \\
        \middlehline
        Settling velocity & $w_{s}$ & $2.5 \times 10^{-4}$ & $m/s$ \\
        Critical bed shear stress for sedimentation & $\tau_{crit,sed}$ & $1000$ & $N/m^2$ \\
        Critical bed shear stress for erosion & $\tau_{crit,ero}$ & $0.2$ & $N/m^2$ \\
        Erosion parameter & $M$ & $1 \times 10^{-4}$ & $kg/m^2/s$ \\
        Specific density & $\rho_{s}$ & $2650$ & $kg/m^3$ \\
        Dry bed density & $\rho_{dry}$ & $1600$ & $kg/m^3$ \\
        \bottomhline
\end{tabular}
\label{sediment}
\end{table*}

Cohesive sediment, i.e. mud, is defined as the mixture of the clay ($<2~\unit{\mu m}$) and silt ($2-63~\unit{\mu m}$) fractions, where its cohesive behaviour is caused mainly  by physico-chemical forces between the clay particles. This cohesive behaviour causes complex processes that influence erosion and deposition of sediments. In the model we distinguish two erosion modes. Above a critical mud content ($p_{m,cr}$) of the bed, sand particles are not in direct contact but are covered by cohesive particles, which limits erosion for both sand and mud \citep{torfs1995,torfs1996}. Below this critical mud content, friction and gravity oppose sediment transport for sand. The critical mud content was chosen to be at a mass fraction of $0.4$, which depends on site specific silt-clay ratios because only the clay fraction is cohesive \citep{mcanally2001, ledden2004}. This value is higher than found in flume experiments ($0.1-0.2$, \citeauthor{torfs1995} \citeyear{torfs1995}; $0.05-0.15$, \citeauthor{torfs1996} \citeyear{torfs1996}; $0.02-0.15$, \citeauthor{mitchener1996} \citeyear{mitchener1996}), but was based on silt-clay ratios of Dutch tidal systems \citep[$0.25-0.5$,][]{ledden2004b}.

When the bed is defined as non-cohesive ($p_{m} < p_{m,cr}$), a traditional sand transport equation was used. Here we chose the Engelund and Hansen transport equation (\citeyear{eh1967}; Eq.~\ref{eq:eh}):
\begin{equation}
\ q_{s} = \frac{0.05 U^{5}} {\sqrt{g} C^{3} \Delta^{2} D_{50}} \label{eq:eh}
\end{equation}
where $q_{s}$~is sediment transport ($\unit{m^3 m^{-1} s^{-1}}$), $U$~is the magnitude of the flow velocity ($\unit{m s^{-1}}$), $\Delta$~is the relative density ($\rho_{s}-\rho_{w})/\rho_{w}$ and $D_{50}$~is the median grain size ($\unit{m}$). This equation does not distinguish between suspended and bedload transport, but considers total transport.

The erosion rate of mud was calculated by the Partheniades-Krone formulation \citep[][Eq.~\ref{eq:PK1}]{partheniades1965}:
\begin{equation}
\ E_{m} = MS(\tau_{cw},\tau_{cr,e}) \label{eq:PK1}
\end{equation}

where $E_m$~is the erosion flux of mud ($\unit{kg m^{-2} s^{-1}}$), $M$~is the erosion parameter ($\unit{kg m^{-2} s^{-1}}$), $S$~is the erosion or depositional step function, $\tau_{cr,e}$~is critical shear stress for erosion ($\unit{N m^{-2}}$), and $\tau_{cw}$~is the maximum bed shear stress due to currents and waves ($\unit{N m^{-2}}$).

When the bed is cohesive ($p_{m}~>~p_{m,cr}$), the mud and sand fluxes are proportional to the mud and sand fraction. The erosion rate of mud is calculated by the Partheniades-Krone formulation (\citeauthor{partheniades1965} \citeyear{partheniades1965}; Eq.~\ref{eq:PK1}) similar to the non-cohesive regime. The erosion rate for sand on the other hand was based on the entrainment of mud, because sand particles are included in the cohesive matrix (Eq.~\ref{eq:EE2}). In this way sand can only be eroded when mud is eroded. Bed load transport was assumed to be zero in the cohesive regime.
\begin{equation}
\ E_{s} = E_{m} \label{eq:EE2}
\end{equation}

Sediment suspended following the Partheniades-Krone formulation was further described by the advection-diffusion equation. Sand and mud behave independently in suspension and segregation will occur with low concentrations \citep{torfs1996}. For simplicity we assumed a constant settling velocity of $0.25~\unit{mm s^{-1}}$ for mud, ignoring that settling velocity depends on flocculation influenced by concentration, residence time, salinity, pH, turbulence and biochemical effects \citep[e.g.,][]{mietta2009}. The settling velocity is typical for fluvial mud \citep[$0.1-0.4~\unit{mm s^{-1}}$,][]{temmerman2003}, which we supply in our default run, and is relatively low for marine mud. Deposition of mud is determined by the concentration, settling velocity and the step function similar to Eq.~\ref{eq:PK1}. However, many studies show that deposition is continuous and a threshold for deposition is therefore absent \citep[short review in][]{sanford2008}. This is approached in the model by setting a very high critical shear stress for sedimentation (Table~\ref{sediment}).

Bed level changes are caused by the divergence of sediment fluxes for bedload and the erosion-deposition difference for suspended sediment. To track the mud and sand fractions in the bed, a bed module was used with a mixed Eulerian-Lagrangian approach \citep{kessel2011,deltares2012} similar to \citet{hir2011} and \citet{sanford2008}. An active Lagrangian layer of $10~\unit{cm}$ was used where sediment exchange occurs with the water column. This active layer had a constant thickness and moved through the vertical framework with bed aggradation and degradationerosion. Below the active layer we used several vertically fixed Eulerian layers to store bed composition in the vertical (Table~\ref{input}). The advantage of Eulerian bed-layers is that artificial mixing by vertically moving layers is prevented. The advantage of a Lagrangian active layer is that the thickness is constant, which is desired because strong bed armouring is prevented and the thickness affects the time scales of the system \citep{deltares2012}.

To speed up morphodynamic calculations the bed level change in each time step, calculated from the divergence of sediment fluxes and the erosion-deposition difference for suspended sediment, was multiplied with a morphological factor of $400$ (Table~\ref{input}). Extensive studies showed reasonable results up to a morphological factor of $1000$, though it is recommended not to go above $400$ \citep[Fig.~\ref{morfacs}]{roelvink2006,wegen2008a}. Using a morphological factor is an efficient way of speeding up long-term morphodynamic calculations that is widely used \citep{ roelvink2006,wegen2008a,hir2015,dam2016}.

When the water level changes during a tidal cycle, flooding and drying of intertidal area occurs. To prevent complicated and time-consuming hydrodynamic calculations with very small water depths a

threshold is set for drying and flooding (Table~\ref{input}). When the water depth is below this threshold the velocity is set to zero. Since the velocity in dry cells is zero, there is no sediment transport in dry cells, even when considerable erosion occurs in a wet cell next to it. Therefore, dry beach and bank erosion was implemented to drive lateral bed lowering. A user-defined factor (Table~\ref{input}) determines the fraction of the erosion flux that is assigned to the adjacent dry cells.

The transverse bed slope effect is a very important parameter for bar dimensions and behaviour in morphological models that is often used as a calibration parameter \citep{schuurman2013}. In estuary models the transverse bed slope effect is often set to be much stronger than the advised default settings to prevent unrealistically steep banks and narrow bars and channels from forming\citep{wegen2012}. The reason for this is unclear but unravelling this is beyond the scope of the present paper so we use settings similar to earlier studies \citep{wegen2012}. We used the transverse bed slope predictor of \citet{koch1980} as extended by \citet{talmon1995}:
\begin{equation}
\ f(\theta) = \alpha \theta^{\beta}
\label{eq:KF}
\end{equation}
where $\theta$ is the shields parameter, $D$ median grain size ($\unit{m}$), $H$ the water depth ($\unit{m}$) and $\alpha=0.2$, much lower than the default of $1.5$ for rivers, and $\beta=0.5$.

\begin{table*}[t]
        \caption{Parameters for processes and numerics}
        \begin{tabular}{ l c c c }
        Parameter & symbol & unit & value \\
        \tophline
        Time step & $dt$ & $\unit{min}$ & $0.3$ \\
        Spin up time at cold start & $-$ & $\unit{min}$ & $1.44 \times 10^{4}$ \\
        Threshold depth drying/flooding & $-$ & $\unit{m}$ & $0.08$ \\
        Min water depth for bed level change & $-$ & $\unit{m}$ & $0.05$ \\
        Erosion adjacent dry cells & $-$ & $-$ & $0.5$ \\
        Morphological factor & $Morfac$ & $-$ & $400$ \\
        Transverse bed slope parameter & $\alpha$ & $-$ & $0.2$ \\
        Transverse bed slope parameter & $\beta$ & $-$ & $0.5$ \\
        Eulerian bed storage layer thickness & $-$ & $\unit{m}$ & $0.1$ \\
        Active layer thickness & $-$ & $\unit{m}$ & $0.1$ \\
        \bottomhline
\end{tabular}
\label{input}
\end{table*}

The Engelund-Hansen transport formulation was chosen because other relations, in particular \citet{vanrijn1993,vanrijn2004,vanrijn2007}, resulted in higher bars and much deeper and straight channels with sudden (up to $90~^\circ$) sharp bends, which would require transverse bed slope parameters that differ by two orders of magnitude from the theoretical value in estuarine settings \citep{wegen2012}. Furthermore, changing bed slope parameters does not fix the channel pattern issues. For long-term morphological modelling Engelund-Hansen produces more realistic morphologies. The disadvantage of our method is that the present code for sand-mud interaction with Engelund-Hansen does not yet incorporate a gradual transition in critical shear stress for erosion between the cohesive and non-cohesive regime. Additionally, mud would ideally erode proportionally with sand in the non-cohesive regime as sand erodes with mud in the cohesive

regime, but this is not yet implemented for Engelund-Hansen and is therefore also not described in our method section. These issues are beyond the scope of the present paper and require further research and model code development.

\subsection{Model schematization}
The modelled domain is $30$~by~$15~\unit{km}$ of which $10$~by~$15~\unit{km}$ is sea area (Fig.~\ref{initialbathy}). The grid is rectilinear with a resolution that varies between $50$~by~$80$ and $125$~by~$230~\unit{m}$. Cell size increases from the initial estuary shape to the sides and offshore to increase resolution in regions of interest and to decrease computation time. The initial bathymetry is in the shape of an idealised funnel-shaped estuary. This exponential shape was also found in previous modelling research \citep{lanzoni2002,canestrelli2008,lanzoni2015} and obtained from field data \citep{savenije2015}. The estuary is $3~\unit{km}$-wide at the mouth and decreases exponentially to a channel of $300~\unit{m}$ wide over $20~\unit{km}$. The bed level linearly increases in elevation from $-2$ at the mouth to $2~\unit{m}$ at the upstream boundary and $2$~to~$3~\unit{m}$ on dry land \citep{wegen2008a}. The sea has a maximum depth of $15~\unit{m}$. \citet{wegen2008a} argued that initial bathymetry does not greatly affect the dynamic equilibrium shape, because dry-cell or bank erosion is allowed in the model and the model will therefore develop a self-formed (alluvial) estuary shape. However, initial shape affects the time needed to form the equilibrium planform shape as well as the size of the ebb delta in the absence of waves and littoral transport, which is the default situation in our idealised estuary. We therefore started with a funnel shape to save calculation time and decrease the size of the ebb tidal delta. The shape is given as:
\begin{equation}
\ W = W_{mouth} e^{\left(\frac{-x}{L_b}\right)}
\label{eq:eshape}
\end{equation}
% \frac{200}{2}+e^{\left(x\frac{\ln \left(\frac{3000}{\left(2\right)}\right)}{20000}\right)}
% \frac{200}{2}+\left(\frac{3200}{2}-\frac{200}{2}\right)^{\frac{x}{20000}}
% 200+\left(3200-200\right)e^{\left(-\frac{x}{3362.6}\right)}
% 3000e^{\left(-\frac{x}{3362.6}\right)}

where $W_{mouth}=3000~\unit{m}$ is the width of the estuary at the mouth , $L_b=3362.6~\unit{m}$ is the e-folding distance over which the width of an exponential channel is reduced by a factor of $e$, and $x$ is distance from the mouth ($\unit{m}$). The shapes of modelled estuaries are characterised by the funnel shape parameter \citep{davies2010} calculated as e-folding length normalised by mouth width at that point in time (Eq.~\ref{eq:fs}). Lower values of the characteristic funnel length indicate stronger funnelling in the sense of more rapid narrowing from the mouth in landward direction. In this way estuary shape is normalised by estuary size.
\begin{equation}
\ S_b = L_b/W_{mouth}
\label{eq:fs}
\end{equation}

\begin{figure*}[t]
\centering
\includegraphics[width=12cm]{initialbathy_width12cm_mdlnr1}
\caption{Initial bathymetry with model boundaries and cross-section (red line) for analysis. Initial depth increases linearly upstream and width decreases exponentially (Eq.~\ref{eq:eshape}). Coordinates are defined at the coastline with the channel centreline and mean sea level (MSL) as origin.}
\label{initialbathy}

\end{figure*}

[revised manuscript text omitted]

\label{hydro}
\end{figure*}
\clearpage

\begin{figure}[t]
\includegraphics[width=8.3cm]{peak_u_ratio_54_109}
\caption{Flood/ebb peak velocity ratio over time for the first $5~\unit{km}$ of the default estuary (run~01) integrated over the total area (black~squares), area above mudflats (brown~circles) and areas above (cyan~triangles) and below (blue~triangles) $-1~\unit{m}$ bed level.}
\label{peak}
\end{figure}

\subsection{Effects of mud supply}
We will now compare other scenarios (run 03, 10 and 09) with the default run (01). In most scenarios mud is accumulating on the flanks of the estuary where the velocities are low and in the upper estuary where it covers a relatively large fraction of the width (Fig.~\ref{mud}f,~g~and~h). Locally, mud accretes on bars that are rather stable (e.g. Fig.~\ref{mud}g~and~h, on the ebb delta). The initiation of mud flats proceeds by the positive feedback identified in the model description: once mud starts settling somewhere, the mud fraction in the bed rapidly increases beyond the critical mud fraction for mud-dominated behaviour. As a consequence, the critical shear stress for sand erosion equals the entrainment threshold of mud (Eq.~\ref{eq:EE2}). The mud-dominated mixed sediment thus becomes more difficult to erode and more rapid aggradation of mud is likely to occur.

Migration rates of channels decrease considerably due to the addition of cohesive material (Fig.~\ref{mud_time}a--h). Bar splitting and merging related to chute cut-offs and avulsion also reduce with increasing mud concentrations. In Fig.~\ref{mud_time}a--d channels move through a cross-section at the mouth through time, though slower for a larger mud supply. For example, a large bar forms in the mouth after about $1100~\unit{yr}$ in the scenario with only sand (Fig.~\ref{mud_time}a) and in the scenario with a mud supply of $50~\unit{mg L^{-1}}$ (Fig.~\ref{mud_time}d). In the run with mud, the bar is covered with mud and becomes fixed while the large bar in the scenario with only sand migrates about $1~\unit{km}$. Absolute bed level changes also indicate that dynamics are decreased with mud input (Fig.~\ref{mud_time}y--II), because there is less bed level change per timestep.

The mudflats have a strong effect on the final shape of the estuary (Fig.~\ref{mud}). Firstly, an increase in fluvial mud input concentration leads to stronger self-confinement of the estuary. By depositing mud on the sides of the estuary, the banks become more stable and limit (further) erosion due to an increased critical shear stress. Self-confinement of estuaries is clearly observed when the models with mud supply are compared to the control run without mud (Fig.~\ref{mud}a). The runs with mud are narrower and have a smaller surface area due to filling of the initial bathymetry, while the sand run has expanded in size. Consequently, the braiding index lowers with increasing mud concentration (Fig.~\ref{mud_length}e--h). In contrast, estuarine surface area continues to increase over time for the control run with only sand (Fig.~\ref{mud_time}q). After the initial rapid change the increase in area and width is linear, driven by dynamic channels and bars and is unhindered by bank stability. This suggests that there is no equilibrium shape under these conditions as is also reflected in the absolute and net bed level change (Fig.~\ref{mud_time}y~and~III). The absolute bed level change does not approach a constant value and the net bed level remains negative, demonstrating the sand-only estuary to be a continuously exporting system.

\begin{figure*}[t]
\includegraphics[width=17cm]{maps_width17cm_runmud_t582}
\caption{Effects of mud supply concentration (run 03, 10, 01 and 09). Left column shows final bathymetry of model runs after $2000~\unit{yr}$ and the right column shows mud fractions in the top layer of the bed. Runs with (a,e) $0~\unit{mg L^{-1}}$, (b,f) $5~\unit{mg L^{-1}}$, (c,g) $20~\unit{mg L^{-1}}$ (default) and (d,h) $50~\unit{mg L^{-1}}$ fluvial mud supply concentration.}
\label{mud}
\end{figure*}

For estuaries with fluvial mud, higher concentrations lead to narrower (Fig.~\ref{mud_length}i--l and Fig.~\ref{mud_time}i--l) and smaller (Fig.~\ref{mud_time}q--t and Fig.~\ref{hypso}) estuaries. Moreover, in some places the width of the estuaries with mud supply is narrower than the initial width, supporting our finding that the initial bathymetry is of limited influence because the system is able to fill and to expand (see methods). Furthermore, tidal bars become higher with increasing mud concentrations, which results in an increased average bed level (Fig.~\ref{mud_length}a--d). Mud is almost nowhere deposited in the channels and does therefore not limit bed erosion by cohesion (Fig.~\ref{hypso}). As a result we infer that the shallower channels in increasingly muddy estuaries mainly result from the decrease in estuary width and concurrent reduction of intertidal area, tidal range and tidal currents (Fig.~\ref{mud_length}).

\begin{figure}[t]
\includegraphics[width=8cm]{hypso_cum_width8cm_run2}

[revised manuscript text omitted]

\begin{figure*}[t]
\includegraphics[width=17cm]{concl_width17cm}
\caption{Most important large-scale morphological parameters after $2000~\unit{yr}$ as a function of the varied boundary conditions: fluvial mud supply concentration, tidal range and fluvial discharge. (a--c) funnel-shape parameter, (d--f) mouth width (in blue colours) and mud flat width (brown colours) at the mouth and (g--i) total area (blue colours) and mud covered area (brown colours). Data indicated in light blue and light brown use a more aggressive masking technique in which high mud flats (higher than $0.5~\unit{m}$ below high water level) are masked from the estuary shape from which area, width and funnel-shape are calculated. Light grey areas indicate models in the transition from estuary to delta. Dark grey indicate models that evolved into a delta.}
\label{concl}
\end{figure*}

\subsection{Comparison to real estuaries}
Model conditions fall within the parameter space of natural estuaries \citep[Fig.~\ref{prandlefig}; Table~\ref{prandletab};][]{prandle2005,leuven2016}. The model has typically larger discharges than the small UK estuaries, but discharge and tidal amplitude falls well within the range of estuaries worldwide.

Several aspects of the bar patterns are further indications that the numerical models reproduce important emergent phenomena of real estuaries. For example, we observe ebb- and flood-dominated channels that are unique for tidal systems \citep{veen1950,ahnert1960}. Typical bar dimensions obtained from the models are in good agreement with natural estuaries from a large dataset \citep{leuven2016}; for example tidal bar length is approximately 7 times the partitioned bar width (maximum bar width devided by barb channels). Furthermore, bar length approximates local width of the estuary. Bars without mud are generally longer and wider for this model study and relative to the local estuary width. Bars in models are also slightly bigger with marine mud supply rather than for fluvial mud supply. The braiding index is strongly related to estuary width as found for natural estuaries \citep{leuven2016} and in agreement with the relation between tendencies to form floodplain in rivers and the resulting relation between channel aspect ratio and bar pattern \citep{kleinhans2010, kleinhans2011, schuurman2016}.

\begin{figure*}[t]
\includegraphics[width=17cm]{bar_length_to_width_and_dependence_width}
\caption{(a) Bar length versus partitioned bar width and (b) bar length against local estuary width. Model results plot in the same range as the data of the natural estuaries \citep{leuven2016}.}
\label{bar_length_to_width_and_dependence_width}
\end{figure*}

[revised manuscript text omitted]

\begin{figure}[t]

\includegraphics[width=8.3cm]{prandle}
\caption{Tidal amplitude plotted against river discharge for real world estuaries and modelled scenarios. Field data is used from \citet{prandle2005} for estuaries in the UK and several other sources for different estuaries over the world. Lines indicate estimations of estuarine length by \citet{prandle2005} of $5, 10$ and $20~\unit{km}$ from left to right.}
\label{prandlefig}
\end{figure}

\begin{table*}[t]
\caption{Ranges of conditions in mixed estuaries at temperate zones \citep{prandle2005} compared to
values for the modelling results.}
\begin{tabular}{ l c c c }
Parameter & Unit & Range & Model \\
        \tophline
        Tidal amplitude & $\unit{m}$ & $1-4$ & $1.5$ \\
        Velocity amplitude & $\unit{m s^{-1}}$ &
$0.5-1.25$ & $0.5-1$ \\
        River discharge & $\unit{m^{3} s^{-1}}$ & $0.25-3000$ & $100$ \\
        Depth at the mouth & $\unit{m}$ & $1-20$ & $2$ \\
        Tidal intrusion length & $\unit{km}$ & $2.5-100$ & $~15$ \\
        Age & $\unit{yr}$ & $100-15000$ & $2000$ \\
        Fall velocity & $\unit{mm s^{-1}}$ & $0.5-5$
        & $0.25 (mud), 41 (sand)$ \\
        \bottomhline
        \end{tabular}
        \label{prandletab}
\end{table*}

[revised manuscript text omitted]

%\disclaimer{?}

\begin{acknowledgements}
This work is part of the PhD project of LB in the project 'Turning the Tide' funded by the Dutch Technology Foundation (TTW) of the Netherlands Organisation for Scientific Research (NWO), Vici-grant 016.140.316/13710 to MK. We would like to thank Pierre Weill and the anonymous reviewer for their contributions to improve the manuscript. Additionally, we are grateful to Mark Macklin, Paul Brewer, Jaco Baas and Alan Davies for a field site visit and discussions on the Dovey estuary and Marco Schrijver of Rijkswaterstaat for field visits and discussion on the Western Scheldt. We acknowledge Bert Jagers of Deltares for help with the sand-mud interaction module of Delft3D, Mick van der Wegen for discussions on long-term modelling, MSc thesis student Samor Wongsoredjo for testing boundary conditions for waves and Anne Baar for comments on an earlier version and discussion. \end{acknowledgements}

\bibliographystyle{copernicus}
\bibliography{paper1}

\end{document}